# STING agonist-loaded, CD47/PD-L1-targeting nanoparticles potentiate antitumor immunity and radiotherapy for glioblastoma

Peng Zhang [1] ✉, Aida Rashidi[1], Junfei Zhao[2,3], Caylee Silvers[1], Hanxiang Wang[1], Brandyn Castro[1], Abby Ellingwood[1], Yu Han[1], Aurora Lopez-Rosas[1], Markella Zannikou[1], Crismita Dmello [1], Rebecca Levine [1], Ting Xiao[1], Alex Cordero[1], Adam M. Sonabend [1], Irina V. Balyasnikova[1], Catalina Lee-Chang [1], Jason Miska [1] ✉ & Maciej S. Lesniak [1] ✉

As a key component of the standard of care for glioblastoma, radiotherapy induces several immune resistance mechanisms, such as upregulation of CD47 and PD-L1. Here, leveraging these radiotherapy-elicited processes, we generate a bridging-lipid nanoparticle (B-LNP) that engages tumor-associated myeloid cells (TAMCs) to glioblastoma cells via anti-CD47/PD-L1 dual ligation. We show that the engager B-LNPs block CD47 and PD-L1 and promote TAMC phagocytic activity. To enhance subsequent T cell recruitment and antitumor responses after tumor engulfment, the B-LNP was encapsulated with diABZI, a non-nucleotidyl agonist for stimulator of interferon genes. In vivo treatment with diABZI-loaded B-LNPs induced a transcriptomic and metabolic switch in TAMCs, turning these immunosuppressive cells into antitumor effectors, which induced T cell infiltration and activation in brain tumors. In preclinical murine models, B-LNP/diABZI administration synergized with radiotherapy to promote brain tumor regression and induce immunological memory against glioma. In summary, our study describes a nanotechnology-based approach that hijacks irradiation-triggered immune checkpoint molecules to boost potent and long-lasting antitumor immunity against glioblastoma.

To date, radiation therapy (RT) remains a mainstream treatment for cancer and is used in treating over 50% of cancer patients[1]. Beyond the genotoxic effects, irradiation-triggered immunogenic cell death and host antitumor immunity have been recognized to play a significant role in the efficacy of RT[2,3]. RT-elicited antitumor immune responses include immunogenic damage-associated molecular pattern (DAMP) signaling, tumor-associated antigen presentation, and activation of tumor-specific effector T cells. Such responses are heavily dependent on the functionality of myeloid cells, an essential component of the innate immune system. However, within the tumor microenvironment, much of the myeloid compartment is programed to be immunosuppressive and tumor-supporting, which greatly blunts the antitumor immunity and limits the therapeutic effects of RT[4–6]. This is particularly true for glioblastoma (GBM), an immunologically "cold" or "quiet" tumor characterized by a lack of sufficient infiltrating effector T cells, but large numbers of immunosuppressive tumor-associated myeloid cells (TAMCs)[7,8]. This remarkable myeloid-rich immune composition[9–12] is a key contributor to the GBM immunosuppression and thus negatively impacts the efficacy of several anti-GBM therapies, including RT, immunotherapy, and chemotherapy[2,13,14]. Consequently,

[1]Department of Neurological Surgery, Lou and Jean Malnati Brain Tumor Institute, Northwestern University Feinberg School of Medicine, Chicago, IL, USA. [2]Program for Mathematical Genomics, Department of Systems Biology, Columbia University, New York, NY, USA. [3]Department of Biomedical Informatics, Columbia University, New York, NY, USA. ✉e-mail: peng@northwestern.edu; jason.miska@northwestern.edu; maciej.lesniak@northwestern.edu

GBM-associated TAMCs have become therapeutic targets of interest in treating brain malignancies.

Targeting TAMCs in cancer treatment has been achieved either through direct depletion or re-education of these cells to promote an antitumor phenotype[4]. Although preclinical studies support the efficacy of the depletion-based approach, this strategy has limitations, including insufficient control of tumor growth and increased incidence of infections[4]. In contrast, approaches for reprogramming TAMCs to promote antitumor immune functions have a substantial upside by providing durable immune responses[5]. Perhaps the most potent approach is the activation of the stimulator of interferon genes (STING) pathway[15,16], which detects cytosolic cyclic dinucleotides and thus triggers downstream transcription of inflammatory genes. Activation of STING in TAMCs induces the production of pro-inflammatory cytokines, including type I interferons (IFNs), which, in turn, stimulates infiltration and activation of effector T cells[16–19]. However, the involvement of STING agonism as part of an attractive immunotherapeutic approach has been hampered by its off-target toxicity. For example, activation of the STING pathway in T cells inhibits T cell proliferation and promotes T cell apoptosis[20,21]. Therefore, new approaches that ensure selective STING activation in specific cell subsets are key to harnessing its potential as part of immunotherapy strategies.

We have previously demonstrated the effectiveness of functionalized lipid nanoparticles (LNPs) for therapeutic delivery to GBM-associated TAMCs in vivo by targeting the programmed death-ligand 1 (PD-L1), which is overexpressed in these cells, particularly when combined with RT[22]. As an encouraging branch of cancer immunotherapy, blocking immune checkpoints regulating adaptive immunity (most commonly effector T cells) has achieved breakthrough in the treatments for several types of cancers. However, GBM patients have yet to benefit from such progress[23] due, in part, to the TAMC-dominant and lymphocyte-diminished tumor microenvironment. Conversely, recent studies have shown that targeting checkpoints that regulate innate immune cells is another powerful method to promote immunotherapy. The first among these checkpoints is CD47, an anti-phagocytic factor that serves as a "don't eat me" signal by tumor cells to escape phagocytic clearance by myeloid cells[24,25]. While anti-GBM responses of anti-CD47 are somewhat limited as a monotherapy[26,27], integrating anti-CD47 mechanisms into a therapeutic platform that coordinately modulates multiple functionalities of TAMCs may prove beneficial.

Towards this objective, we developed a bridging-lipid nanoparticle (B-LNP) with anti-CD47/PD-L1 dual-targeting capability to simultaneously block innate (CD47) and effector (PD-L1) checkpoint molecules while also delivering the STING agonist, diABZI, to TAMCs in GBM. Not only does this platform block checkpoint molecules, but it also serves as a bridge forcing an interaction between TAMCs (via PD-L1) and tumors (via CD47), thus promoting the phagocytic capacity and antigen presentation. In addition, the B-LNPs are also loaded with a STING agonist, thus promoting STING signaling in TAMCs, increasing subsequent T cell recruitment (Fig. 1). Our work suggests that B-LNPs combined with STING agonists may serve as a powerful therapeutic modality to promote antitumor immunity and complement existing standard of care treatments for GBM.

## Results

### B-LNP promotes TAMC phagocytosis of irradiated glioma cells

The highly expressed anti-phagocytic factor CD47 in tumors is well-studied[24,25] and recapitulated in our orthotopic syngeneic glioma cell line CT-2A[28]. We first analyzed how RT affects the CD47 expression in GBM. Figure 2a shows that RT upregulated CD47 expression in CT-2A, suggesting CD47 overexpression as tumor adaptation to therapeutic stress. However, along with upregulation of CD47, irradiation also greatly increased cell surface expression of calreticulin (CRT), a dominant pro-phagocytic molecule that initiates phagocytic clearance of dying tumor cells[29] (Fig. 2b, Supplementary Fig. 1a). Such concurrent induction of pro-/anti-phagocytic signals by irradiation was also demonstrable in vivo (Supplementary Fig. 2). To leverage these RT-elicited processes and break the balance between pro-/anti-phagocytic signals that glioma cells use to evade phagocytosis, we engineered a B-LNP that is functionalized with (i) anti-CD47 antibody (αCD47), allowing for targeting and blocking of overexpressed CD47 in gliomas, as well as (ii) anti-PD-L1 antibody (αPD-L1) for binding and engaging of TAMCs to glioma cells (Fig. 2c). PD-L1 was used as the myeloid-targeting moiety as it is highly expressed in CD45+ CD11b+ TAMCs over CD45+CD11b- tumor-infiltrating lymphocytes (TILs) and tumor cells (Fig. 2d, Supplementary Fig. 1b). Conversely, RT-triggered over-expression of CD47 was observed in CT-2A tumor but not much in TAMCs and TILs (Fig. 2d, Supplementary Fig. 1b). Therefore, this platform should facilitate interactions between TAMCs and glioma which may be enhanced by tumor radiation.

The B-LNP has a small and uniform particle size of around 90 nm as determined by dynamic light scattering (DLS) and transmission electron microscopy (TEM) (Supplementary Fig. 3a), and a slightly negative surface charge (−4.41 mV). In vitro, B-LNP efficiently bound to the surface of CT-2A, which blocked the anti-phagocytic factor CD47 (Fig. 2e, Supplementary Fig. 1c). This binding did not interfere with the TAMC targeting capability of B-LNP through PD-L1 ligation, as indicated by results from a competition assay using an excess amount of free blocking antibodies (Fig. 2f, Supplementary Fig. 1d). Such PD-L1 ligation also efficiently blocked the highly expressed PD-L1 in TAMCs

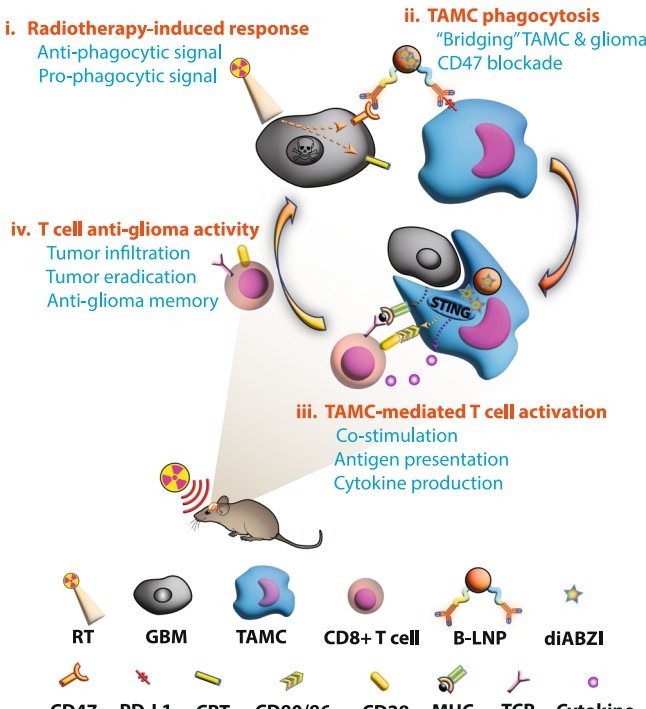

**Fig. 1 | Schematic representation of B-LNP-mediated therapeutic hijacking of TAMCs to boost antitumor immunity and potentiate radiotherapy for GBM.** **i**, Radiotherapy (RT) induces overexpression of pro-phagocytic signal calreticulin (CRT) and anti-phagocytic molecule CD47 in glioma cells. **ii**, B-LNPs were engineered to "bridge" TAMCs and GBM through CD47 and PD-L1 dual-ligation, block both checkpoint molecules, and promote phagocytosis of tumor cells. **iii**, In addition to acting as an engager, B-LNPs also encapsulate STING agonist to promote type I interferon responses in TAMCs, which trigger infiltration and activation of tumor-antigen specific T cells. **iv**, Reprogrammed TAMCs promote T cell-mediated antitumor responses, leading to durable tumor eradication. TCR, T cell receptor; MHC, major histocompatibility complex. The figure was generated using Adobe Illustrator. The mouse illustration was sourced and adapted from Scidraw.io (doi.org/10.5281/zenodo.3925921).

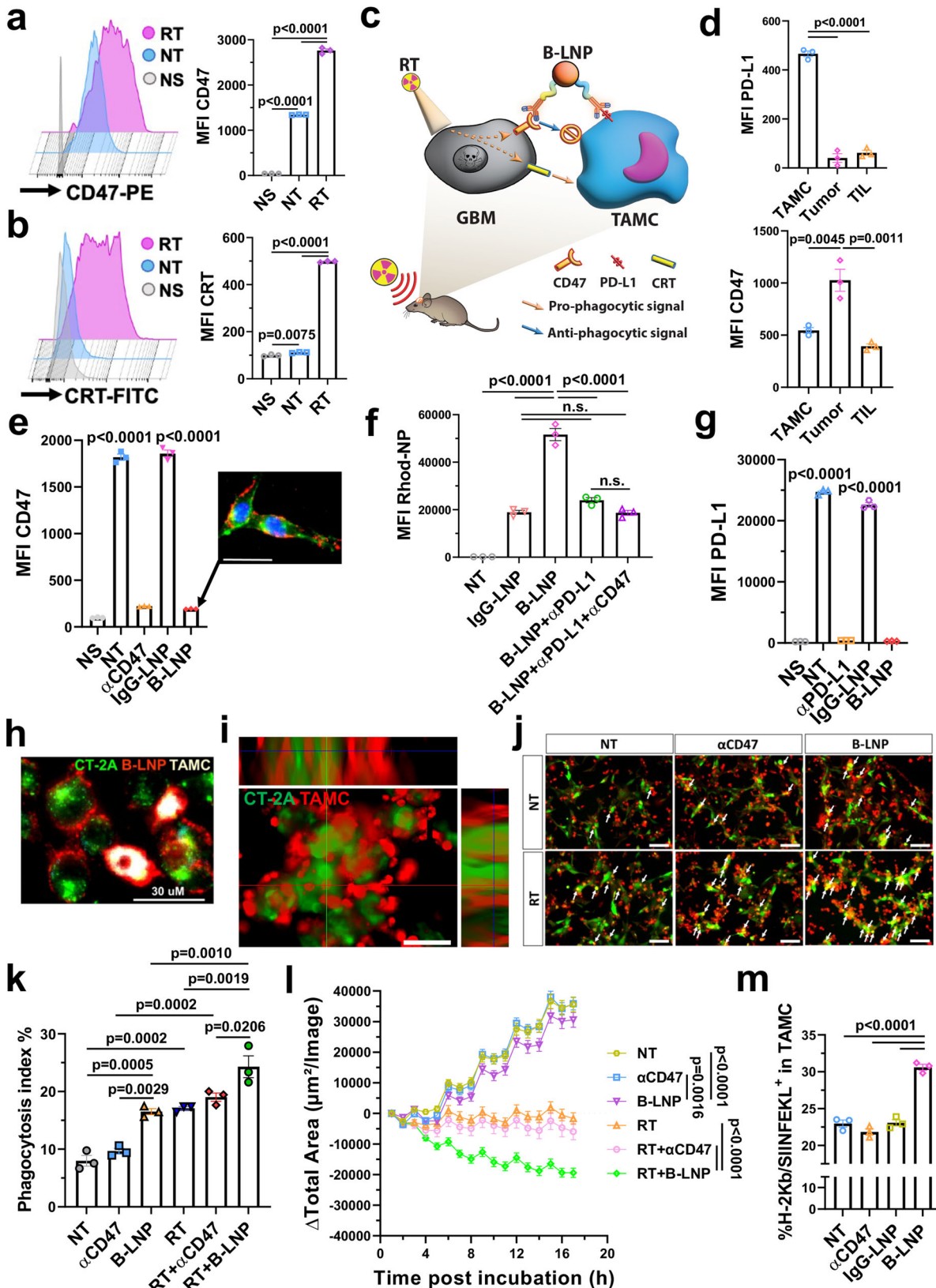

(Fig. 2g, Supplementary Fig. 1e). B-LNPs demonstrated similar binding kinetics to both TAMCs and gliomas (Supplementary Fig. 3b), and the cellular binding of B-LNPs promptly induced TAMC-glioma coupling as revealed by fluorescence microscopy (Fig. 2h) and Z-stack imaging (Supplementary Fig. 4). The ability of B-LNP to form TAMC-glioma cell clusters was also observed using Z-stack imaging through the co-culture of TAMCs and glioma cells in low cell adherence plates (Fig. 2i). These results demonstrate B-LNP promoted TAMC-glioma interactions through a "bridging" effect via a glioma←CD47-LNP-PD-L1→TAMC axis. As a result, treatment with B-LNP significantly increased TAMC phagocytic activity of glioma (Fig. 2j, k), and significantly enhanced clearance of irradiated glioma cells (Fig. 2l). Using CT-2A

**Fig. 2 | B-LNPs engage TAMC and promote phagocytosis. a, b** Flow cytometric quantification of radiotherapy (RT)-triggered overexpression of CD47 (**a**) and calreticulin (CRT) (**b**) in CT-2A glioma cells 72 h after 9 Gy of irradiation. Data are shown as mean fluorescence intensity (MFI). NT, non-treated; NS, unstained control. $n = 3$ independent samples. **c** Schematic presentation of the "bridging" effect of B-LNP to engage TAMC and block the checkpoint molecules CD47 and PD-L1. The scheme was generated using Adobe Illustrator. The mouse illustration was sourced and adapted from Scidraw.io (doi.org/10.5281/zenodo.3925921). **d** Flow cytometric quantification (MFI) of PD-L1 and CD47 expression in different subsets of cells in CT-2A-bearing brains 72 h after brain-focused radiotherapy. $n = 3$ mice. **e** Flow cytometric quantification (MFI) of CD47 expression in CT-2A cells treated with αCD47 antibody at 25 μg/ml. $n = 3$ independent samples. Figure insert indicates cell membrane binding of Rhod-tagged B-LNP (red) in CT-2A cells (green) with nuclear staining (blue). Scale bar, 30 μm. **f** Binding efficiency of Rhod-tagged B-LNP to CT-2A-associated TAMCs. $n = 3$ donor mice. **g** Flow cytometric quantification (MFI) of PD-L1 expression in CT-2A-associated TAMCs treated with αPD-L1 antibody at

25 μg/ml. $n = 3$ biological replicates. **h** Cellular distribution of Rhod-tagged B-LNP (red) in a co-culture of CT-2A (green) and TAMC (gray) for 30 min. Scale bar, 30 μm. The experiment was carried out independently three times. **i** Z-stack microscopic image of co-cultured irradiated CT-2A (green) and TAMC (red) in an ultra-low attachment plate. Scale bar, 100 μm. The experiment was carried out independently three times. **j, k** TAMC (red) phagocytosis of CT-2A (green) +/- RT treated with αCD47 antibody (free or B-LNP-conjugated form) at 10 μg/ml for 4 h at 37 °C. Arrows indicate the phagocytic cells. Phagocytic index was determined by fluorescence microscopy. $n = 3$ donor mice. Scale bar, 100 μm. **l,** Kinetics of TAMC phagocytic clearance of CT-2A were measured by IncuCyte in a co-culture of CT-2A-GFP and TAMCs. $n = 16$ views by IncuCyte. **m** Flow cytometric quantification of OVA$_{257-264}$ (SIINFEKL) peptide bound to H-2K$^b$ in TAMCs 24 h after co-cultured with CT-2A-OVA. $n = 3$ biological replicates. One-way ANOVA with Tukey's multiple comparisons test was used in all figure panels. The data are presented as mean +/- SEM. Representative dot plots of flow cytometric analysis are provided in Supplementary Fig. 1. Source data are provided as a Source Data file.

overexpressing ovalbumin (CT-2A-OVA) (Supplementary Fig. 5) as a model system, we encountered that the B-LNP-stimulated phagocytosis of glioma cells led to an enhanced presentation of tumor-associated antigens by TAMCs (Fig. 2m, Supplementary Fig. 1f).

## B-LNP-mediated STING activation remodels transcriptomic, metabolic, and phenotypic features of TAMC

Then we aimed to test if STING agonism could reprogram immunosuppressive TAMCs to pro-inflammatory cells. In vitro generated TAMCs with a highly immunosuppressive phenotype (Supplementary Fig. 6) were treated with diABZI, a recently developed synthetic STING agonist[30]. Treatment of diABZI increased expression of pro-inflammatory genes in TAMCs, including type I IFNs, C-C motif chemokine ligand 5 (CCL5), and C-X-C motif chemokine ligand 10 (CXCL10) (Supplementary Fig. 7a), each of which is known for T cell recruitment and activation[31,32]. Our flow cytometry data also suggest that diABZI-treated TAMCs expressed reduced levels of immunosuppressive factors arginase 1 (ARG1) and CD206, and higher levels of activation factor CD86 (Supplementary Fig. 7b). To enable a specific STING activation in TAMCs but not in T cells to avoid the off-target toxicity (Supplementary Fig. 7c), B-LNP was used to encapsulate diABZI (referred to as B-LNP/diABZI) through a thin-film rehydration method[22,33]. The formulated B-LNP/diABZI demonstrated a small size of 93 nm (Supplementary Fig. 3c), satisfactory stability (Supplementary Fig. 3d), and a high TAMC-targeting capability over other subsets of glioma-infiltrating immune cells (Supplementary Fig. 8).

To uncover how TAMCs were changed by such TAMC-targeted therapeutic intervention, a single-cell RNA sequencing (scRNA-seq) analysis was performed using cell suspensions prepared from the brains of glioma-bearing mice that received RT +/- B-LNP/diABZI treatments. An integrated analysis was conducted to compare transcriptional profiles of different cell subsets within the brain tumor microenvironment post-RT or RT + B-LNP/diABZI combination therapy (referred to as Combo) (Fig. 3a). As depicted by Uniform Manifold Approximation and Projection (UMAP) and cell composition analysis (Fig. 3b), TAMCs represented the most predominant immune cell population in the murine glioma microenvironment, among which the percentage of monocytes was largely increased by the combination therapy. As excepted, Combo treatment caused a transcriptional change in TAMCs, as an increased subset of TAMCs that demonstrated pro-inflammatory phenotype was observed (sub-cluster 1, 14.6% v.s. 28.7%) (Fig. 3c, Supplementary Fig. 9a, Supplementary Dataset 1). Unbiased analysis of the transcriptomic shift in TAMCs revealed that Combo-treated TAMCs highly expressed myeloid cell activation and interferon-related pro-inflammatory genes (such as Saa3, Slpi, Nos2, Rsad2, Slfn4, Cmpk2, Cxcl10, and Ifit1) not only in a greater proportion of cells but also to a higher degree of expression (Fig. 3d). We further investigated the enriched TAMC functions through gene set

enrichment analysis. Pathways associated with pro-inflammatory and immunostimulatory activities were identified as the top upregulated ones in TAMCs post-Combo treatment as compared to RT alone (Fig. 3e, Supplementary Dataset 2). Furthermore, a polarization analysis through a panel of selected genes also indicates an anti-inflammatory to pro-inflammatory switch of TAMCs post-Combo therapy, as defined by a set of highly expressed pro-inflammatory markers (Cxcl9, Cxcl10, Ccl5, Irf7, CD40, CD86, Nos2, Tnf), while anti-inflammatory genes were expressed at a significantly lower level (Arg1, Mrc1) (Fig. 3f). The switched phenotypic feature of TAMCs was also confirmed by flow cytometry analysis of glioma-bearing animals (Fig. 3g). It is worth noting that microglia, a prominent brain intrinsic myeloid population, were also positively affected by the Combo treatment (Supplementary Fig. 10), which is consistent with the nanoparticle uptake efficiency of microglia (Supplementary Fig. 8).

The effect of B-LNP/diABZI treatment was also demonstrated by changes of the metabolic phenotype of TAMCs. Metabolic reprogramming of myeloid cells is critical to their identity as either tumor suppressors or those which promote tumor growth[34]. In order to address this, we performed magnetic bead-based isolation of TAMCs from RT +/- B-LNP/diABZI tumors, followed by untargeted metabolite profiling (Fig. 4a). Analysis of untargeted metabolomics revealed four distinct metabolic pathways altered by Combo treatment: purine, pyrimidine, alanine, and arginine metabolism (Fig. 4b). By cross-referencing the genetic changes from scRNA-seq analysis with our metabolomics datasets, we found that the Combo therapy induced a significant switch in arginine metabolism from the generation of polyamines towards the activation of inducible nitric oxide synthase (iNOS)-derived metabolites (Fig. 4c). Specifically, we found a significant increase in iNOS-derived metabolites citrulline and L-argininosuccinate ($p < 0.05$) (Fig. 4d), and conversely, a decrease in arginase-derived metabolites ornithine and urea (Fig. 4e). We also found significant reduction in immunosuppressive polyamines spermidine and spermine ($p < 0.01$) (Fig. 4f). This comports with the analysis of our scRNA-seq datasets (Fig. 3f). Our data collectively indicate that the Combo treatment reshaped the transcriptomic and metabolic profiles of TAMCs, thus increasing their pro-inflammatory functions in the GBM microenvironment.

## Therapeutic reprogramming of TAMC enhances CD8$^+$ T cell infiltration and activation in murine glioma models

To demonstrate the effects of nanoparticle-mediated TAMC reprogramming on the architecture of glioma immune landscape, particularly antitumor adaptive immunity (Fig. 5a), transcriptomic changes of T cells were also evaluated through scRNA-seq analysis. An unbiased gene expression analysis of T cell clusters indicates an overexpression of a panel of interferon-stimulated genes, such as Isg15, Usp18, Irf7, Bst2, Zbp1, and Rtp4 (Fig. 5b), and a gene enrichment analysis of the

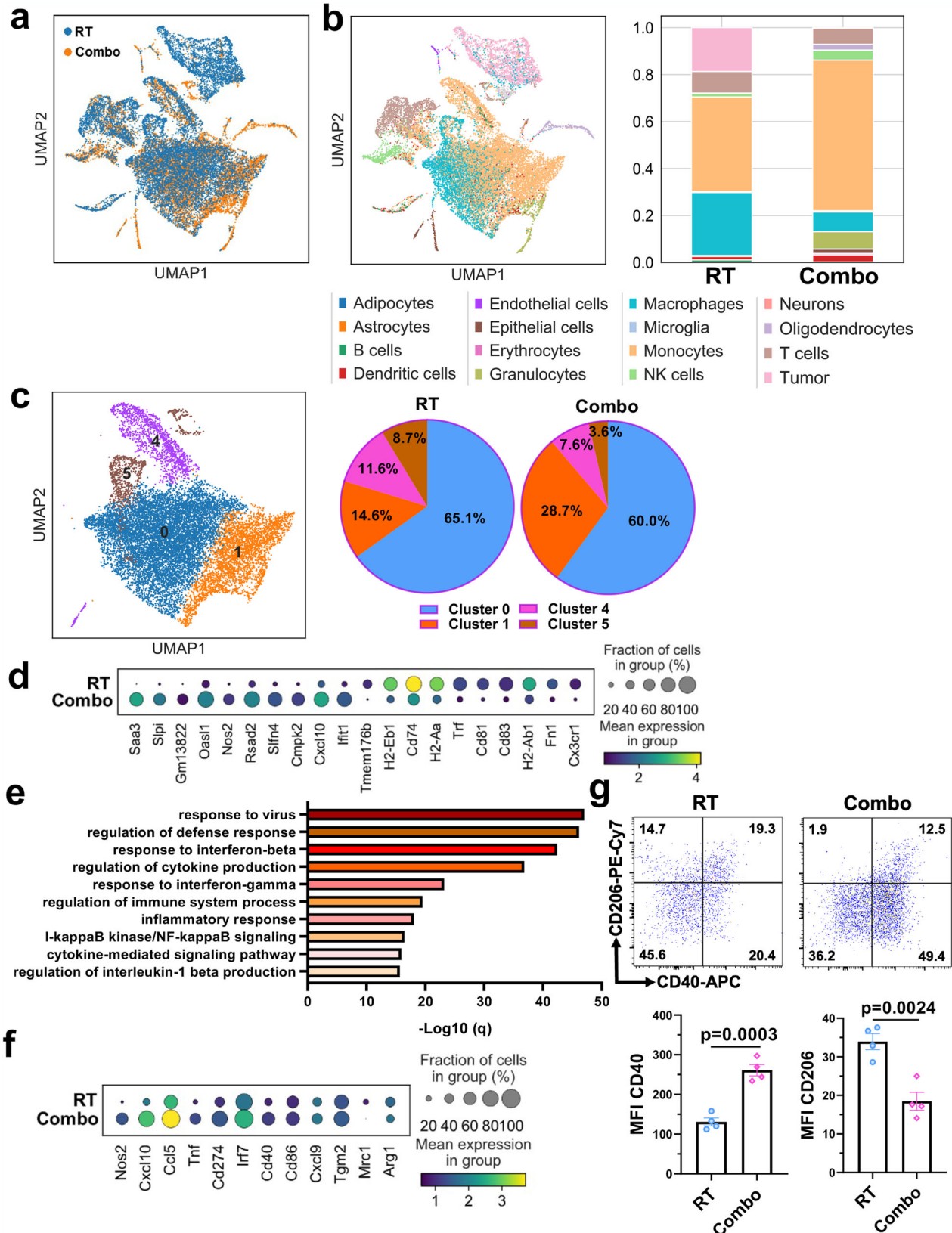

top upregulated genes in T cells post-Combo treatment further reveals an increased proliferation and activation status of antitumor T cells, as well as T cell responses to IFNs (Fig. 5c, Supplementary Dataset 3). These results suggest the impact of B-LNP-mediated pro-inflammatory reprogramming of TAMCs on antitumor activity of T cells in the brain. As a result, there was a significantly increased fraction of antitumor

T cells with an activated status and enhanced cytotoxic activities, as characterized by highly expressed CD69, IFN-γ, and granzyme B (GzmB), but lower expression of programmed cell death protein 1 (PD-1) (Fig. 5d). Notably, T cells were also observed to mostly express high Ki67 among different subsets of cells (Fig. 5e), indicating a proliferative status of T cells post-Combo treatment. In addition,

**Fig. 3 | B-LNP-mediated targeted delivery of diABZI remodels transcriptomic and phenotypic features of TAMC. a**–**f** Single-cell RNA sequencing analysis of brain tumors from CT-2A-bearing C57 mice received radiotherapy (RT) or RT + B-LNP/diABZI combination therapy (referred to as Combo) (n = 5 mice pooled per sample). **a** UMAP plots of the clustering of cells collected from brain tumors as color-coded by treatments (RT, blue v.s. Combo, orange). **b** Cell composition analysis of brain tumors after RT or Combo treatments. **c** Sub-cluster analysis of TAMCs. Gene enrichment analysis and the full list of the upregulated genes in sub-cluster 1 are provided in Supplementary Fig. 9a and Supplementary Dataset 1. **d** Unbiased expression analysis indicating top upregulated and downregulated genes in TAMCs post-Combo treatment as compared to RT alone. **e** GO analysis of top upregulated pathways in TAMCs post-Combo treatment. q-values were determined using the Benjamini-Hochberg procedure to account for multiple testing. The full gene list is provided in Supplementary Dataset 2. **f** Expression analysis of representative genes in related to TAMC pro-inflammatory and anti-inflammatory activities. **g** Flow cytometric analysis of CD40 and CD206 expression in TAMCs from CT-2A-bearing brains of C57 mice. n = 4 mice. Statistics were determined by two-sided Student's t-test. The data are presented as mean +/- SEM. Source data are provided as a Source Data file.

strengthened T cell-glioma interactions through chemokine signaling, such as CCR5-CCL7/CCL8, CCR2-CCL7/CCL8/CCL11, CXCR3-CXCL9/CCL19, and CXCR6-CXCL16, were also revealed by a ligand-receptor analysis (Supplementary Fig. 9b). T cell clonotyping also reveals an increase in T cell clonal diversity post-Combo treatment (Fig. 5f), which is likely due to the enhanced clonal expansion of glioma-reactive T cells driven by improved TAMC presentation of tumor-associated antigens.

Along with the results of scRNA-seq analysis, enhanced trafficking of CD8[+] T cells to brain tumors was observed in CT-2A-bearing mice that received Combo treatments. About 6-fold increase in infiltration as well as activation of adaptively transferred CD45.1[+] CD8[+] T cells was observed in brain tumors of the recipient mice post-nanoparticle treatment (Fig. 6a). The B-LNP-mediated CD8[+] T cell infiltration was mainly observed in the brain tumor site (Fig. 6b). To understand the underlying mechanism of T cell homing to brain tumors, we evaluated local and peripheral pro-inflammatory cytokines through Meso Scale Discovery (MSD) multiplex analysis. Our results indicate increased T cell-recruiting/activating cytokines CXCL10 and CCL2 both in serum and the brain tumor milieu (Fig. 6c). Increased CCL3 and IL30 were also observed, while CXCL2 and IL33 were not significantly changed (Supplementary Fig. 11). The impact of B-LNP on antitumor T cells was further evaluated using free drug combinations (equivalent dose of free diABZI + αPD-L1 + αCD47, referred to as "Cocktail") as a control. Our results indicated that the "Cocktail" treatment was not able to induce the same effects as B-LNP/diABZI (Fig. 6d), neither were the drug-free B-LNPs (Supplementary Fig. 12), suggesting the importance of using B-LNPs for TAMC-targeted STING activation. It is also worth noting that increased CD8[+] T cell exhaustion was observed post-B-LNP/diABZI or Cocktail treatment (Supplementary Fig. 13), indicating a potential benefit of combining immune checkpoint inhibitors with our therapy to reverse the post-therapy T cell exhaustion.

To further determine the potential of our TAMC reprogramming approach for clinical translation, B-LNP/diABZI was synthesized using anti-human antibodies and tested ex vivo using GBM patient samples. GBM cells and GBM-infiltrating immune cells were harvested from clinical tumor specimens, and mixed in a co-culture with peripheral T cells isolated from paired blood samples from the same patients (Fig. 7a). Flow cytometry data suggested that nanoparticle treatment induced activation of both peripheral and tumoral CD8[+] T cells, as identified by increased expression of CD69 and CD25 (Fig. 7b, c), but did not stimulate PD-l expression (Supplementary Fig. 14a). Such T cell activation is likely due to the nanoparticle-enabled TAMC activation (Supplementary Fig. 14b). It was also determined that the non-adherent immune cells were significantly clustered in the co-culture treated by B-LNP/diABZI (Supplementary Fig. 14c), and the enhanced cell-cell clustering interactions may indicate a rapid expansion and antigen-specific activation status of immune cells[35,36], which is consistent with the results from flow cytometry. Activation of TAMC and CD8[+] T cell by nanoparticle treatment was similarly observed in another batch of patient-derived cells (Supplementary Fig. 14d). In addition to endogenous T cells, we also explored the capability of using B-LNP treatment to potentiate migration of chimeric antigen receptor (CAR) T cells from blood circulation to GBM. Excitingly, one

dose of B-LNP/diABZI injection induced an over 4-fold increase in CAR T cell infiltration to brain tumors, which led to a reduction of tumor burden by 75% (Supplementary Fig. 15).

## Nano-therapeutic potentiates antitumor effects of standard of care and provides long-term survival in gliomas

The findings that B-LNP/diABZI treatment promotes RT-triggered antitumor immune responses motivated us to examine antitumor efficacy when combined with standard of care. CT-2A is an aggressive, immunologically "cold", and therapy-resistant murine brain tumor model[37], to which RT only showed marginal therapeutic effect (median survival: control, 20 days; RT, 24 days) (Fig. 8a). However, intracranial administration of B-LNP/diABZI improved the antitumor efficacy of RT, leading to an extended animal survival with 60% of glioma-bearing animals being cured of CT-2A tumors and observed to be long-term survivors (LTS) with no indication of tumor burden or therapy-caused brain toxicity. As a comparison, the treatment with RT + diABZI-free B-LNP only showed marginal antitumor effects in CT-2A-bearing mice (Supplementary Fig. 16).

To gain more insight into the underlying mechanisms of treatment effects, we further tested the treatments in different glioma-bearing transgenic mice models. First, we implanted CT-2A into C57 STING[Gt] mice, in which the mice cannot produce type I interferons upon STING activation, whereas the tumor cells maintain STING responses. The therapeutic benefits of combination treatment were not observed in the STING[Gt] mice (Fig. 8b). Next, C57 Rag1[−/−] mice without mature lymphocytes[38] were used to determine the role of adaptive immunity in the treatment effects. As indicated by Fig. 8c, antitumor effects of RT + B-LNP/diABZI combination therapy were largely impaired in the absence of adaptive immune cells, and adoptive transfer of CD8[+] T cells partially rescued the survival benefit as observed in wildtype mice. Along with these results, immuno-profiling analysis of LTS mice indicated a greatly increased TIL population in the tumor-free brains of LTS as compared to control mice without tumor implantation, among which an increased amount of CD8[+] T cells, particularly memory CD8[+] T cells, was observed (Fig. 8d). The effector CD8[+] T cells in LTS mice showed an activated phenotype with high expression of IFN-γ (Fig. 8e).

In addition to RT alone, we also investigated therapeutic outcomes when combining B-LNP/diABZI with RT plus concomitant adjuvant chemotherapeutic temozolomide (TMZ), known as the "Stupp protocol", which is routinely used as standard of care in treating newly-diagnosed GBM[39,40]. RT + TMZ provided modest but measurable survival benefits to animal subjects (median survival: control, 27.5 days; RT + TMZ, 36 days). Free drugs and antibodies administered in combination modestly prolonged animal survival as compared to RT + TMZ alone (median survival: 50 days). However, the combination of B-LNP/diABZI with RT + TMZ most significantly improved the survival benefit (median survival: 114 days) (Supplementary Fig. 17).

## Reprogramming of TAMC through systemic delivery route enhances anti-glioma effects

To foster a potential clinical translation of our nano-therapeutic platform, we also tested the possibility of administrating the

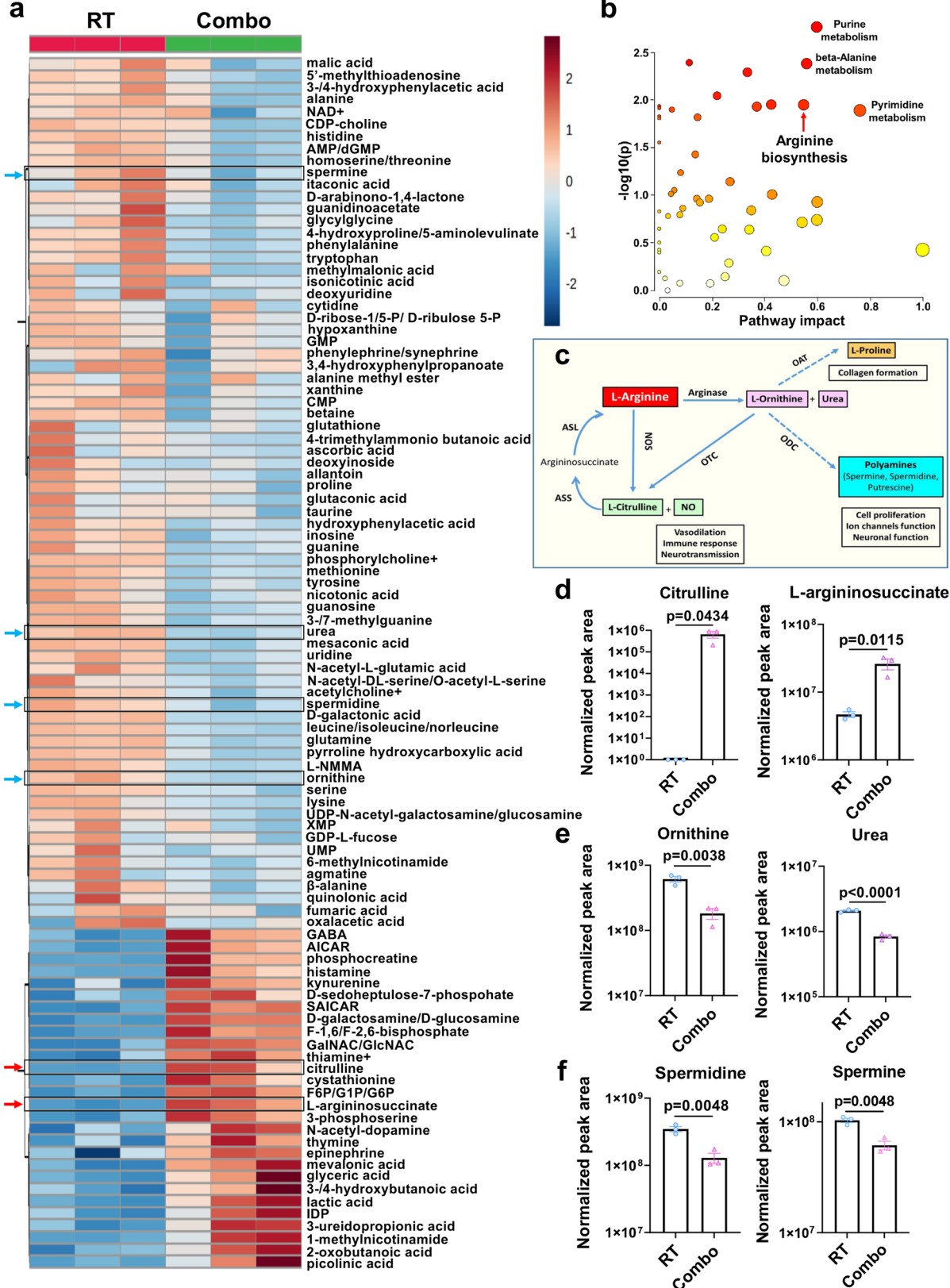

**Fig. 4 | B-LNP/diABZI treatment reprograms metabolic features of TAMC.** Bulk metabolomic analysis of TAMCs from CT-2A-bearing brains of C57 mice received radiotherapy (RT) or RT + B-LNP/diABZI combination therapy (referred to as Combo). Unbiased metabolomics (**a**) and metabolic pathway analysis (**b**) of TAMCs from brain tumors. **c** Schematic illustration of the arginine metabolic pathway. LC-MS/MS analysis of iNOS (**d**), ARG1 (**e**), and ornithine decarboxylase (ODC) (**f**)-derived metabolites in TAMCs. *n* = 3 samples (TAMCs collected from 5 CT-2A-bearing mice were pooled for each sample; in total, 15 mice per treatment group). Statistics were determined by two-sided Student's t-test. The data are presented as mean + /- SEM. Source data are provided as a Source Data file.

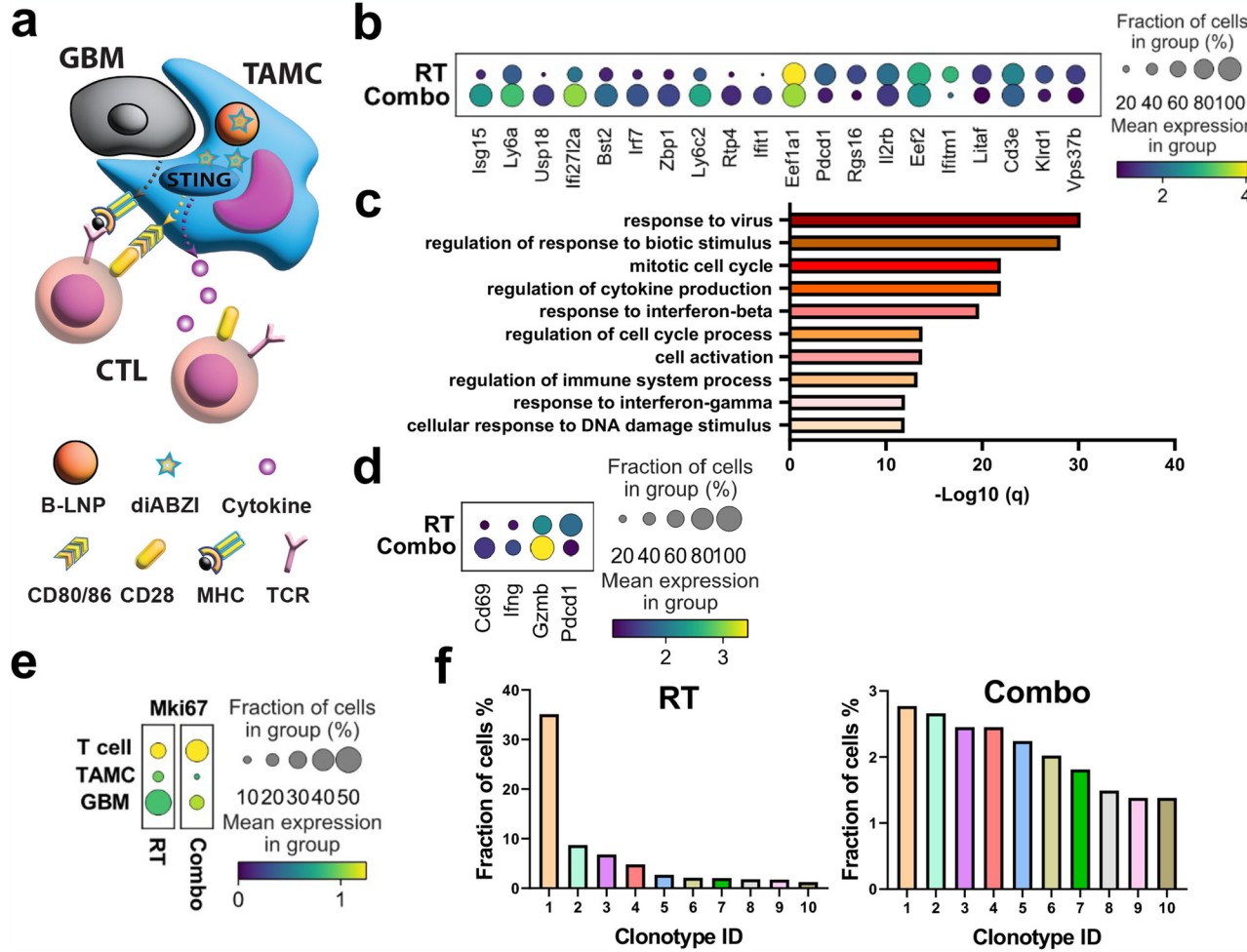

**Fig. 5 | Nanoparticle-mediated TAMC reprogramming induces transcriptomic changes of T cells in murine gliomas. a** Schematic representation of the interaction between B-LNP/diABZI-reprogrammed TAMC and T cell. The scheme was generated using Adobe Illustrator. **b**–**f** Single-cell RNA sequencing analysis of T cells from CT-2A brain tumors in C57 mice received radiotherapy (RT) or RT + B-LNP/diABZI combination therapy (referred to as Combo) (n = 5 mice pooled per sample). **b** Unbiased differential expression analysis indicating top upregulated and downregulated genes in T cells post-Combo treatment as compared to RT alone. **c** GO analysis of top upregulated pathways in T cells post-Combo treatment. q-values were determined using the Benjamini-Hochberg procedure to account for multiple testing. The full gene list is provided in Supplementary Dataset 3. **d** Expression analysis of representative genes in related to T cell activation. **e** Differential expression analysis of Mki67 expression in T cell, TAMC, and tumor cell. **f** T cell clonotype analysis.

immunostimulatory nanoparticles through a systemic delivery approach. Towards this objective, diABZI was formulated by a TAMC-targeting LNP with anti-PD-L1 functionalization (P-LNP)[22], in which the anti-CD47 mechanism was not incorporated to avoid the unexpected binding to peripheral erythrocytes which also highly express CD47[41]. The P-LNP demonstrated high efficiency in homing to brain and TAMC targeting, as revealed by live animal imaging and immunofluorescence staining (Supplementary Fig. 18). Similar to the results of intracranial administration, a regimen combining fractionated RT with intravenously injected P-LNP/diABZI, but not free drugs, led to enhanced CD8+ T cell tumor infiltration (Supplementary Fig. 19) and animal survival in CT-2A glioma model, eradicating engrafted tumor from 70% of the glioma-bearing mice (Fig. 9a). Indicating successful immunologic memory, all the survived mice effectively rejected rechallenged CT-2A tumor in the opposite hemisphere without any sign of tumor growth and tissue toxicity (Fig. 9b, c). After 100 days following tumor rechallenge, the brains of the surviving mice were evaluated for CD8+ T cell status. Immunoprofiling analysis suggested that there was an increase in CD8+ T cells demonstrating a memory phenotype in the brains of LTS mice, and the effector CD8+ T cells also highly expressed IFN-γ but not GzmB

(Fig. 9d). This memory-like CD8+ T cell status may largely contribute to the long-term immunological memory against gliomas.

Besides CT-2A, we also evaluated if our nano-therapeutic treatment could amplify the RT effects in tumors with known genetic drivers. To this end we utilized PDGF+PTEN-/- (PVPF8) murine glioma cells, which were generated through intracranial injection of PDGF-IRES-Cre retrovirus into PtenFlox B6 mice[42,43]. This model recapitulates the genetic alterations, histological features, and tumor-associated immunosuppressive phenotype as seen in human GBM. As indicated by Fig. 10a, PVPF8 glioma exhibits a diffuse infiltration feature and pseudopalisading necrosis. In addition, PVPF8 represents an "immune desert" tumor model with extremely low immune cell infiltration. However, two doses of P-LNP/diABZI effectively boosted antitumor immunity, including pro-inflammatory TAMCs as well as activated T cells (Fig. 10b, Supplementary Fig. 20). Importantly, intravenous administration of P-LNP/diABZI vastly improved the antitumor effects of RT in such animals bearing PVPF8 glioma (Fig. 10c). These data, all together, may indicate a great potential of using systemically injectable TAMC-reprogramming nano-therapeutics to boost anti-glioma immunity as a unique approach to potentiate the antitumor effectiveness.

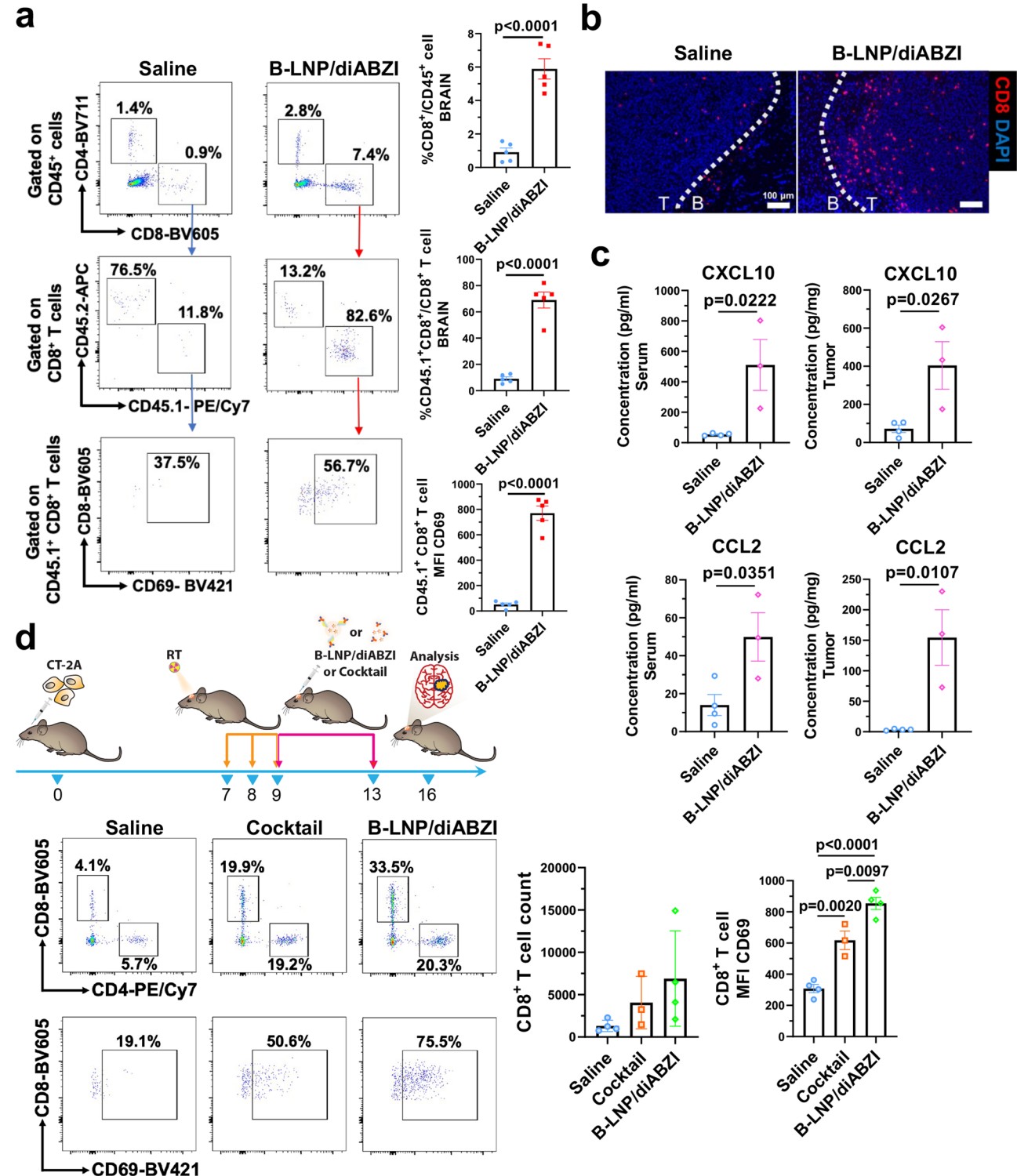

**Fig. 6 | Therapeutic reprogramming of TAMC enhances CD8⁺ T cell infiltration and activation in murine gliomas. a** CD45.1⁺ CD8⁺ T cells were adoptively transferred through intravenous administration to CT-2A tumor-bearing C57 mice received radiotherapy (RT) followed by saline or B-LNP/diABZI (0.25 mg/kg diABZI) treatment through an intracranially implanted cannula. CD45.1⁺ cells were analyzed by flow cytometry in tumor-bearing brains (*n* = 5 mice). T cell activation status was assessed by CD69 expression. A representative animal for each group is shown. **b** CD8⁺ T cell brain localization was demonstrated by immunofluorescence staining. Dotted line indicates the border of normal brain (B) and tumor site (T). Scale bar, 100 μm. The experiment was carried out twice independently. **c** MSD multiplex cytokine analysis of serum and brain tumors collected from CT-2A-bearing

C57 mice received RT + saline (*n* = 4 mice) or B-LNP/diABZI (*n* = 3 mice). **d** CT-2A-bearing mice were treated by RT + saline, free diABZI + αPD-L1 + αCD47 (Cocktail), or B-LNP/diABZI. T cells were analyzed by flow cytometry in tumor-bearing brains (*n* = 3 or 4 mice/group). T cell activation status was assessed by expression of CD69. The scheme was generated using Microsoft PowerPoint and Adobe Illustrator. The mouse (doi.org/10.5281/zenodo.3925921) and syringe (10.5281/zenodo.4152947) illustrations were sourced and adapted from Scidraw.io. Statistics were determined by two-sided Student's t-test (in **a**, **c**) or one-way ANOVA with Tukey's multiple comparisons test (in **d**). The data are presented as mean +/- SEM. Source data are provided as a Source Data file.

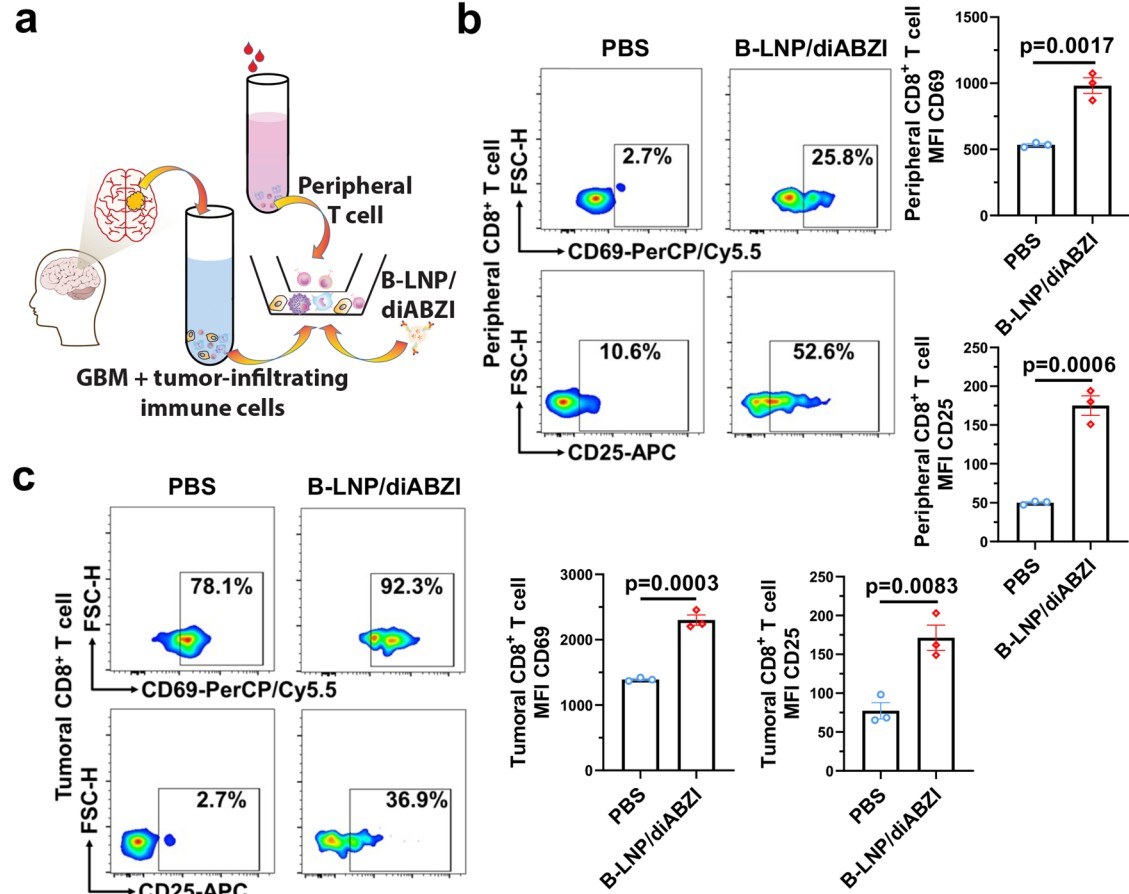

**Fig. 7 | Anti-human CD47/PD-L1 functionalized B-LNP/diABZI activates GBM patient-derived T cells. a** Schematic representation of ex vivo experiments using clinical specimens from GBM patients. The scheme was generated using Adobe Illustrator. The human brain illustration was sourced and adapted from Scidraw.io (doi.org/10.5281/zenodo.3925925, doi.org/10.5281/zenodo.3926065). Ex vivo samples were treated with PBS or anti-human CD47/PD-L1 functionalized B-LNP/diABZI at diABZI concentration of 100 nM. Flow cytometric analysis of activation status of peripheral (**b**) and tumoral (**c**) CD8[+] T cells collected from clinical specimens of GBM case NU02747. *n* = 3 independent formulation samples. Statistics were determined by two-sided Student's t-test. The data are presented as mean +/- SEM. Source data are provided as a Source Data file.

## Discussion

In this study, we demonstrated a design of B-LNP with dual functionality: (i) reinforces TAMC phagocytic and antigen-presenting activity; and (ii) induces STING activation and pro-inflammatory responses in TAMCs. This nano-therapeutic enhanced radiotherapy by hijacking the irradiation-inducible immune checkpoints CD47 and PD-L1 to both target and inhibit immune suppression, while harnessing multiple TAMC activities to amplify the irradiation-triggered antitumor immune responses. Therefore, B-LNP represents a unique therapeutic and delivery platform that leverages the tumor-eradicating v.s. tumor-supporting mechanisms induced by radiotherapy to promote antitumor immunity.

One major advantage of B-LNP is that the nanocarrier, rather than solely serving as a drug delivery vehicle, by itself also exerts biological functionality, promoting TAMC phagocytic activity. Phagocytosis represents a key antitumor function of myeloid cells that includes the "sampling", processing, and presentation of tumor-associated antigens, which induces subsequent activation of adaptive immunity[44,45]. Our B-LNP promoted TAMC phagocytosis of tumor cells through multifaceted mechanisms. First of all, the dual-targeting capability of B-LNP enables it to serve as an engager to bridge TAMCs and GBM cells, reinforcing the pro-phagocytic signals in TAMC-GBM communication via enhanced ligand-receptor interactions. Second, the surface-decorated anti-CD47 molecules on B-LNP block the anti-phagocytic signal overexpressed in GBM

cells and thus overcomes the barrier that GBM uses to escape myeloid clearance. Particularly, irradiation highly stimulates GBM expression of calreticulin but concurrently upregulates anti-phagocytic molecule CD47 to balance this pro-phagocytic signal[29]. These facts lend us theoretical support to use B-LNP to break the balance employed by GBM to evade immune recognition and harness the antitumor functionality of TAMCs for promoted tumor engulfment and tumor-associated antigen presentation. Notably, the fact that CD47 is highly overexpressed in GBM over normal tissue[28] in the brain indicates a lower risk of off-target binding-induced toxicity to normal cells.

Besides stimulating TAMC phagocytosis, another key objective of our therapy is to increase effector T cell antitumor functions through TAMC-specific STING pathway activation. Poor tumor infiltration and impaired antitumor activity of effector T cells are key parameters of immunologically "cold" tumors and have been associated with low therapy response rate[3]. Although compelling evidence has demonstrated the ability of STING agonists to trigger and augment pro-inflammatory immune responses and stimulate infiltration and activation of effector T cells[16–19], its off-target activation and toxicity in T cells remain a concern[20,21]. Building on its high effectiveness in targeting TAMCs, B-LNP as a drug carrier efficiently delivers diABZI to GBM-associated TAMCs and reprograms such cells to exert immunostimulating activity by the production of pro-inflammatory and T cell recruiting cytokines. The facts that increased T cell infiltration

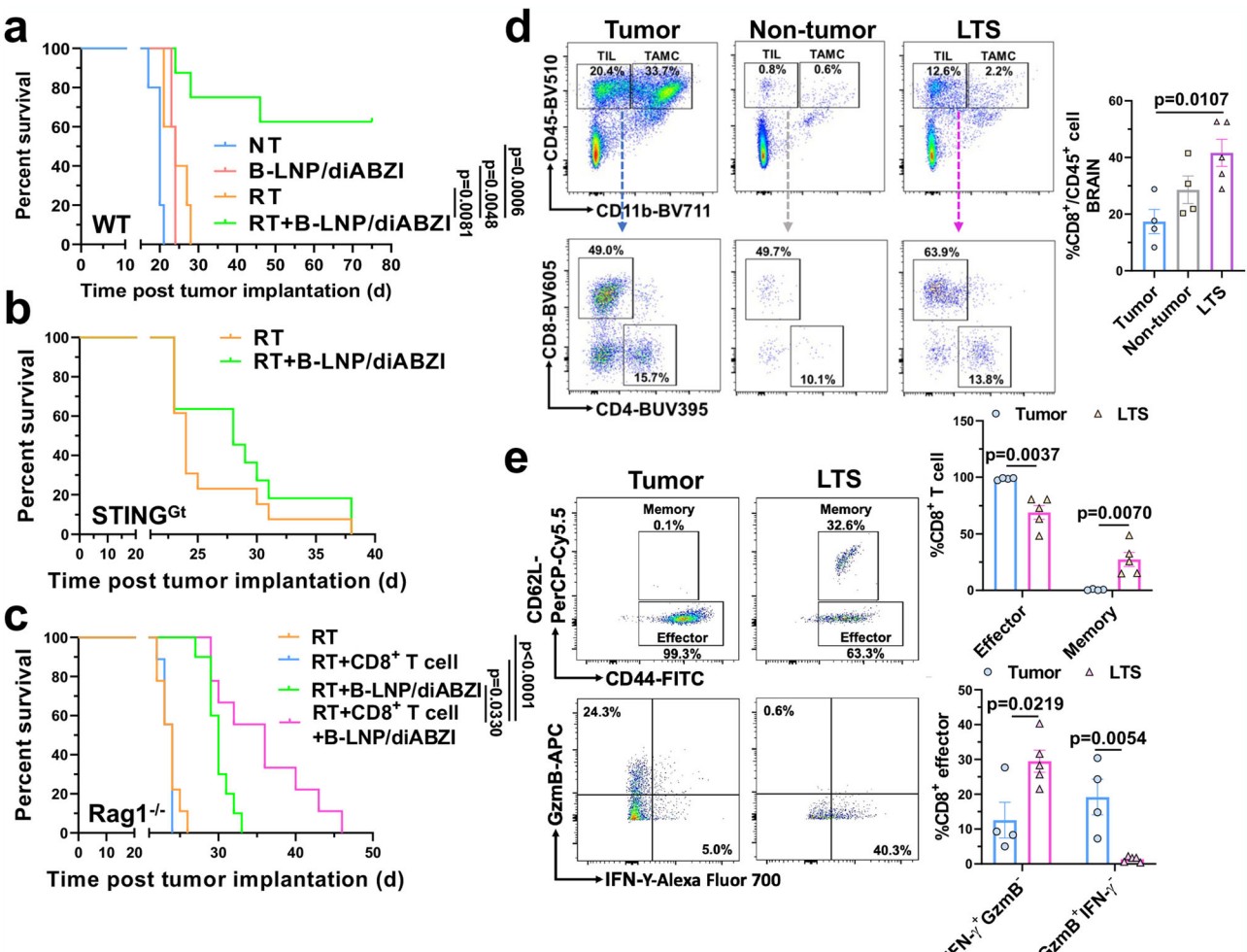

**Fig. 8 | B-LNP/diABZI treatment potentiates the antitumor effects of radio-therapy in glioma-bearing mice. a–c** Survival curves of C57 wildtype (WT) (**a**), STING[Gt] (**b**), and Rag1[-/-] (**c**) mice received intracranial implantation of 5×10⁴ CT-2A glioma cells and two administrations of saline or B-LNP/diABZI (0.25 mg/kg diABZI) through an intracranially implanted cannula. Selected groups of mice received radiotherapy (RT, 3 Gy×3) as monotherapy or combination therapy starting from the seventh day post-tumor implantation. In (**c**) selected groups of mice also received two doses of adoptively transferred CD8⁺ T cells (5×10⁶ per dose) through intravenous injection. Statistics were determined by Log-rank method with *p* values adjusted by Bonferroni correction. **d–e** Freshly dissected brains from long-term survivor (LTS), age-matched tumor-bearing mice, and age-matched non-tumor control mice were analyzed by flow cytometry for immune composition (**d**) and CD8⁺ T cell phenotypes (**e**). *n* = 4-5 mice. TIL, tumor-infiltrating lymphocyte. Statistics were determined by one-way ANOVA with Tukey's multiple comparisons test (in **d**) or two-sided Student's t-test (in **e**). The data are presented as mean +/- SEM. Source data are provided as a Source Data file.

and activation were observed in glioma-bearing mice post-B-LNP/diABZI treatment, while neither free drug combinations nor drug-free B-LNP were able to show the same potency, highlight the importance of B-LNP-mediated coordinate modulation of multiple TAMC anti-tumor mechanisms over solely promoting phagocytosis or activating STING pathways.

In addition to the induction of T cell infiltration, B-LNP-mediated TAMC reprogramming also creates a favorable microenvironment for activation, expansion, and persistence of T cells in brain tumors. Our results indicate that B-LNP treatments induced TAMC expression of co-stimulatory factors (CD40, CD80, CD86) and enhanced tumor-associated antigen presentation, and, importantly, impaired TAMC immunosuppressive activities. It was worth noting that, as supported by flow cytometry, scRNA-seq, and bulk metabolomics, TAMCs were turned from Arg1⁺ into iNOS⁺ TAMCs. Considering the potently immunosuppressive properties of polyamine metabolites[46], these results suggest the B-LNP treatment rewires arginine metabolism in murine models of brain tumors towards a less immunosuppressive phenotype, breaking the metabolic barriers that immunosuppressive TAMCs used to impair effector T cell proliferation and activation in GBM[46].

These facts altogether indicate that B-LNP remodels TAMC functionality from immunosuppressive/tumor-supportive to immunostimulatory/tumor-eradicating, which (i) triggers brain tumor infiltration of effector T cells, and (ii) creates a more T cell-favorable tumor microenvironment. Supporting this notion, administration of B-LNP/diABZI improves the antitumor effects of brain-focused irradiation, eradicating tumor from over 60% of glioma-bearing mice, which, however, was not achieved in trans-genic mice models lacking STING or those lacking adaptive immune cells[38]. The diminished therapeutic effects of combination treatment in these mouse models suggest that STING activation in the host immune system, rather than tumor itself, is the major mechanism of the treatment effects, and that the TAMC reprogramming-mediated antitumor functions of adaptive immunity, especially CD8⁺ T cells, play an important role in the treatment effects, which are strongly supported by our in vivo scRNA-seq data. Notably, the B-LNP treatments also elicited long-term immune responses against glioma, as evidenced by the prevention of tumor rechallenge and the largely increased number of memory CD8⁺ T cells residing in the brain. These results altogether strongly suggest the effectiveness of our TAMC-reprogramming nanoparticle as

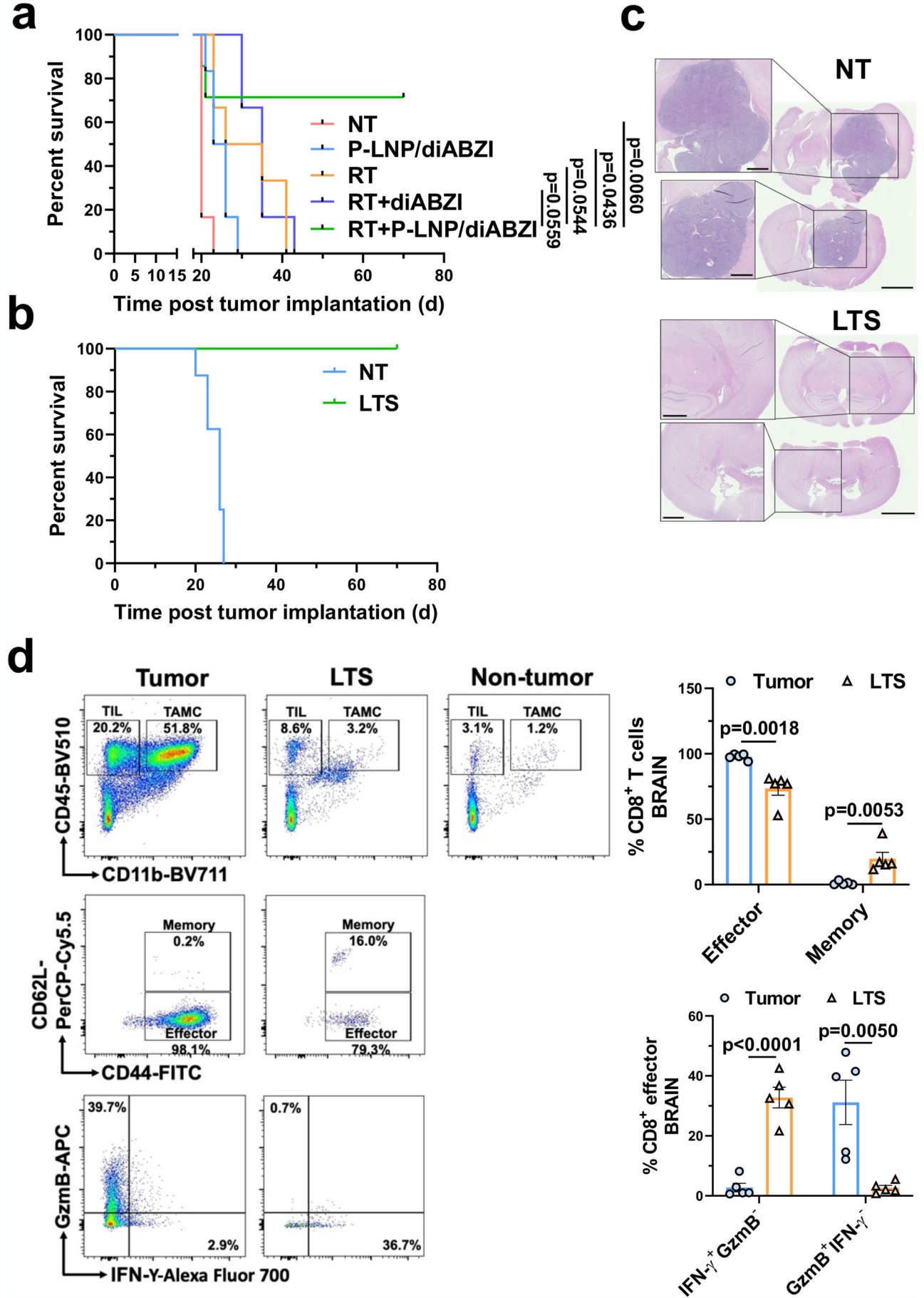

**Fig. 9 | Systemic nanoparticle delivery of diABZI potentiates the anti-glioma effects of radiotherapy.** C57 mice received intracranial implantation of $7.5 \times 10^4$ CT-2A cells and three administrations of saline, free diABZI, or P-LNP/diABZI (2.5 mg/kg diABZI) through intravenous injections. Selected groups of mice received radiotherapy (RT, 3 Gy×3) as monotherapy or combination therapy. **a** Survival curves of CT-2A-bearing C57 mice received different treatments. $n = 6–7$ mice per group. Survival curves were compared using Log-rank test for those with proportional hazards or Renyi's test for those with crossing hazards with $p$ values adjusted by Bonferroni correction. **b** 180 d after tumor implantation, long-term survivor (LTS) mice were rechallenged with $7.5 \times 10^4$ CT-2A cells in the opposite hemisphere of the initial tumor injection site. The animal survival was compared to age-matched tumor-bearing control mice (NT). **c** Tumor burden in control and LTS mice were evaluated through H&E staining. Scale bar, 2.5 mm; insert scale bar, 1 mm. **d** 100 d post-rechallenge, freshly dissected brains from LTS, age-matched tumor-bearing mice, and age-matched non-tumor control mice were analyzed by flow cytometry for immune composition and CD8$^+$ T cell phenotypes. $n = 5$ mice. Statistics were determined by two-sided Student's t-test. The data are presented as mean +/- SEM. Source data are provided as a Source Data file.

a combination therapy approach to enhance the radiotherapy effects for GBM therapy. It is worth noting that, besides TAMC, another key component of the myeloid compartment in the brain, microglia, could also take nano-therapeutics and be turned into a pro-inflammatory feature. This observation has added merit to our therapy in the context of GBM and needs to be further investigated.

Aiming to establish a translation-ready formulation, the clinical transition potential of the immunostimulatory B-LNP was thoroughly assessed by various aspects. First, the effectiveness of the nano-therapeutics was further evaluated under a more clinically relevant circumstance when combined with RT plus chemotherapeutic TMZ, which is routinely used in the post-surgical treatment of newly diagnosed GBM patients. Considering the clinical feasibility of locally delivering therapeutics through the convection-enhanced delivery for GBM patients[47,48], our nano-therapeutics hold a potential to be used as an adjuvant to the standard of care therapy in the context of primary and/or recurrent GBM. Furthermore, the fact that B-LNP/diABZI therapy could improve CAR T cell migration to brain tumors has opened another door to potentiate current immunotherapeutic strategies. Since inefficient CAR T cell penetration to brain tumors remains a major obstacle to the success of cell therapy for GBM patients[49], our results may suggest B-LNP as a unique and effective solution to address the current limitations. Importantly, besides intracranial administration, our systemically injectable nanoparticle formulation was also effective in preclinical animal models of the disease in terms of brain homing and antitumor effects. Most excitingly, our therapy turned PVPF8, a murine model that recapitulates multiple genetic and histologic features of human GBM[42,43], from an immunological "desert" into an immunologically "hot" tumor resulting in extended animal survival. Lastly, a human-ready lipid nanoparticle formulation was evaluated through the use of clinical specimens, confirming its efficacy in activating antitumor immunity in human GBM. As lipid nanoparticles are well-known as a clinically successful nanoparticle platform, a rapid translation of our nano-therapeutic for clinical evaluation is anticipated.

In conclusion, this work establishes a therapeutic approach that represents a feasible and translatable therapeutic for specific TAMC targeting and reprogramming. As part of an integrated multimodal therapy, it holds tremendous potential to coordinately engage, harness, and modulate diverse TAMC functionalities to augment the antitumor effects of standard of care treatments. The demonstrated ability to remodel the immunosuppressive and "cold" GBM microenvironment may surmount major hurdles for effective treatment of GBM and can be readily applied to address the unmet clinical need for improving GBM patient outcomes.

## Methods
### Animals
C57BL/6 (RRID: IMSR_JAX:000664), CD45.1 (RRID: IMSR_JAX: 002014), Rag1 deficient (RRID: IMSR_JAX:002216), OT-I (RRID:IMSR_JAX:003831), and STING$^{Gt}$ (RRID: IMSR_JAX:017537) mice were all purchased from the Jackson Laboratory. Animals were bred and housed in a standard barrier animal facility at Northwestern University with the light cycle of 14:10, ambient temperature at 22 °C, and relative humidity range between 30-70%. Experimental animals were mixed-gender and randomly assigned to different treatment groups at 6 to 8 weeks old. Experimental and control animals were co-housed. All animal experiments were performed in full compliance with animal protocols approved by the Northwestern University Institutional Animal Care and Use Committee (IACUC).

### Human samples
All human samples (tumor and peripheral blood) were collected by the Nervous System Tumor Bank at Northwestern University (NSTB) under the Institutional Review Board (IRB) protocol N° STU00095863 and STU00202003. Patients presenting for a neurosurgical procedure for the treatment of a nervous system tumor and meeting the inclusion criteria were consented for participation in the tumor tissue bank at the time of consent for the surgical procedure. The study was conducted in accordance with U.S. Common Rule of ethical standards. All cases were reviewed by a board-certified neuropathologist. The diagnostic criteria were based off the WHO 2021 classification of tumors.

### Reagents
1,2-dioleoyl-sn-glycero-3-phosphocholine (DOPC), 1′,3′-bis[1,2-dioleoyl-sn-glycero-3-phospho]-glycerol (18:1 cardiolipin), 1,2-distearoyl-sn-glycero-3-phosphoethanolamine-N-[methoxy(polyethylene glycol)−2000] (DSPE-PEG$_{2000}$), 1,2-distearoyl-sn-glycero-3-phosphoethanolamine-N-[maleimide(polyethylene glycol)−2000] (DSPE-PEG$_{2000}$-maleimide), 1,2-distearoyl-sn-glycero-3-phosphoethanolamine-N-(Cyanine 5.5) (18:0 Cy5.5 PE), and L-α-Phosphatidylethanolamine-N-(lissamine rhodamine B sulfonyl) (Rhod-PE) were all purchased from Avanti Polar Lipids. Cholesterol and 2-iminothiolane hydrochloride were purchased from Sigma Aldrich. diABZI STING agonist (Compound 3) was purchased from Selleckchem. Temozolomide was purchased from Cayman Chemical. InVivoMAb antibodies including anti-mouse PD-L1 (B7-H1, clone 10F.9G2), anti-mouse CD47 (clone MIAP301), anti-human PD-L1 (B7-H1, clone 29E.2A3), and anti-human CD47 (clone B6.H12) were all purchased from BioXCell.

### Cell lines
CT-2A murine glioma cells were obtained as a gift from Dr. Tom Seyfried at Boston College. PVPF8 cells were generated by and obtained from Dr. Adam Sonabend at Northwestern University. CT-2A cells were maintained in DMEM (Corning) with 10% fetal bovine serum (FBS, HyClone) and 1% penicillin/streptomycin (Invitrogen). PVPF8 cells were maintained in high glucose DMEM with L-glutamine (Gibco) supplemented with 0.5% FBS, 1% penicillin/streptomycin, 1% N2 supplement (Gibco), platelet-derived growth factor-AA (PDGF-AA, Peprotech, 10 ng/ml), and murine fibroblast growth factor-basic (FGF, Peprotech, 10 ng/ml). All cell lines were incubated at 37 °C in 5% CO$_2$ humidified atmosphere.

### Genetically modified cell lines
For CT-2A cell line overexpressing ovalbumin (CT-2A-OVA), CT-2A cells were transfected with the pAc-Neo-OVA plasmid (Addgene1) using Lipofectamine 2000 reagent (Fisher). The transfected CT-2A cells were selected for 10 days in 200 μg/ml of G-418 (Sigma). The overexpression of OVA was confirmed by Western blot using ovalbumin antibody (Novus Biologicals, 1:500 dilution) (Supplementary Fig. 5a). CT-2A-

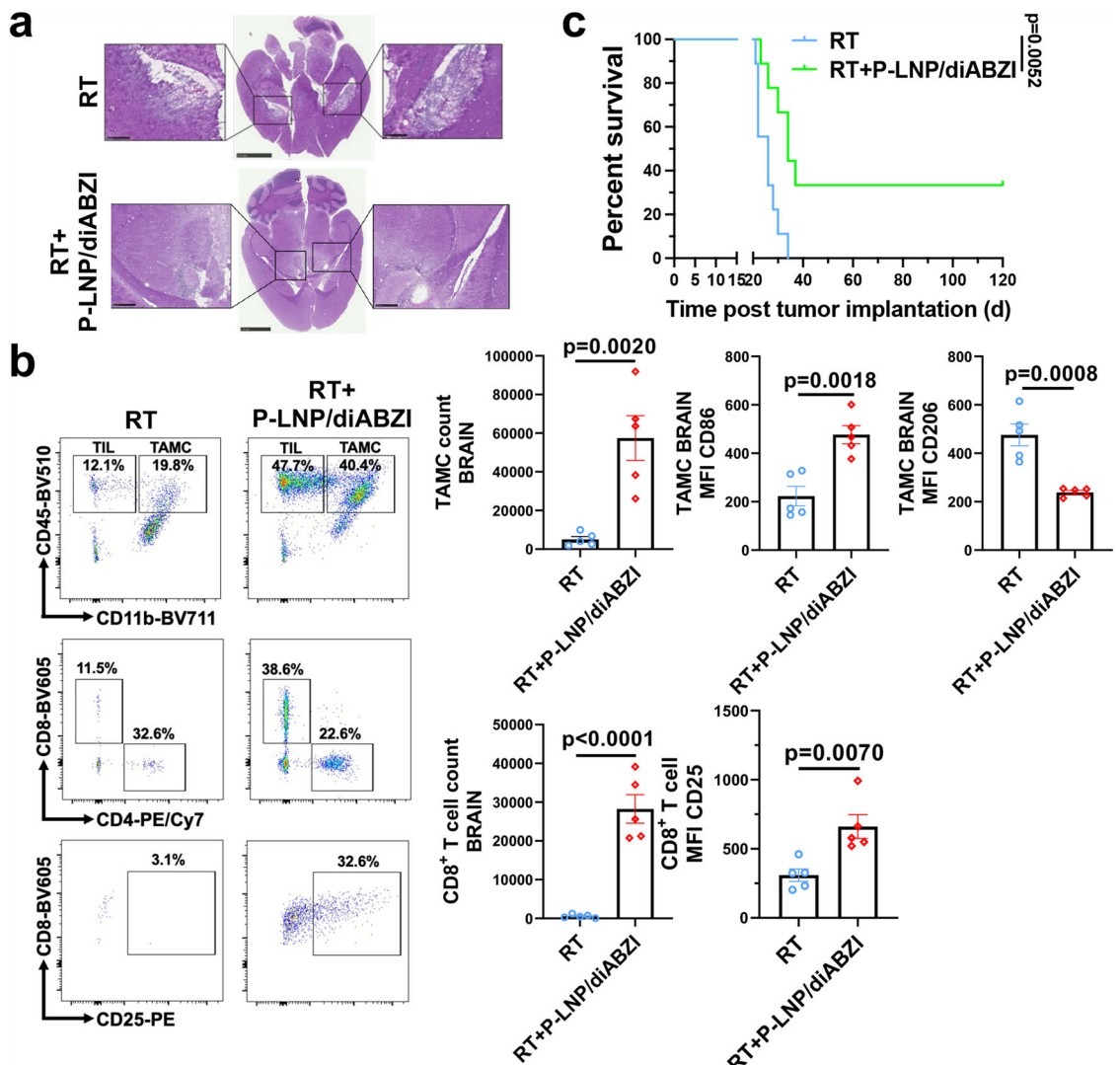

**Fig. 10 | Nanoparticle treatment reshapes immune microenvironment and potentiates the therapeutic effects of radiotherapy in PVPF8 murine gliomas.** C57 mice received intracranial implantation of $7.5 \times 10^4$ PVPF8 cells, radiotherapy (RT, 3 Gy×3) and two administrations of saline or P-LNP/diABZI (2.5 mg/kg diABZI) through intravenous injections. **a** Histology of the brains was evaluated by H&E staining. Scale bar, 2.5 mm; inset scale bar, 500 μm. **b** Freshly dissected brains were analyzed by flow cytometry for immune composition, TAMC, and CD8+ T cell phenotypes. $n = 5$ mice. Statistics were determined by two-sided Student's t-test; data are presented as mean +/- SEM. Representative dot plots of flow cytometric analysis are provided in Supplementary Fig. 20. **c** Survival curves of the PVPF8 tumor-bearing mice received RT (3 Gy×3) and two administrations of saline or P-LNP/diABZI (2.5 mg/kg diABZI) through intravenous injections. $n = 9$ mice. Statistics were determined by Log-rank method. Source data are provided as a Source Data file.

OVA cells were maintained in DMEM with the addition of 200 μg/ml of G-418 for selection pressure. For CT-2A cell line overexpressing green fluorescent protein (CT-2A-GFP), pLVX-IRES-ZsGreen1 plasmid (Takara Bio) and 4th generation Lenti-X™ Packaging Single Shots (Takara Bio) were used to produce lentiviral particles in HEK 293/17 cells. 24 and 48 h later the cell supernatants were collected and concentrated using a Lenti X concentrator (Takara Bio). CT-2A cells were transduced with viral concentrate in the presence of 1 μg/ml polybrene (Sigma). CT-2A cells containing ZsGreen1 protein were expanded and sorted on the basis of fluorescence (Supplementary Fig. 5b).

**Synthesis and characterization of B-LNP/diABZI**

B-LNPs were synthesized through a thin-film rehydration method as described in[22]. Briefly, DOPC, cholesterol, 18:1 cardiolipin, DSPE-PEG$_{2000}$, and DSPE-PEG$_{2000}$-maleimide were mixed at a molar ratio of 56:29:10:3:2 in chloroform in a glass vial, followed by the addition of diABZI at a molar ratio of 20:1 (total lipids: diABZI). In case fluorescent tag was needed, 0.5% (molar ratio) Rhod-PE or 18:0 Cy5.5 PE was added

into the lipid mixture. A thin film of mixed lipids/diABZI was obtained by removing the organic solvent through a gentle nitrogen stream and then dried in vacuum for 4 h, followed by being hydrated in DPBS (Sigma) for 1 h, resuspended by brief vortex for 30 s, and homogenized using a probe sonicator (Active Motif). Drug-free nanoparticles were prepared similarly without adding diABZI to lipids. αPD-L1 and αCD47 (anti-mouse or anti-human) antibodies (BioXCell) were mixed at 1:1 molar ratio, then incubated with 2-iminothiolane hydrochloride at a molar ratio of 1:10 in DPBS (pH 8.0, 4 mM EDTA) at room temperature for 1 h, followed by purification using Amicon Ultra-15 centrifugal filter units (MWCO: 10 kDa, Millipore). The modified antibodies were incubated with LNPs in DPBS (pH 7.0) at a weight ratio of 8.7:1 (total lipids to antibody) at 4 °C for overnight, then purified by centrifugation. P-LNP/diABZI was prepared following the same procedure with αPD-L1 decoration. The particle size distribution and surface charge of B-LNP/diABZI was determined using a Zetasizer Nano ZSP (Malvern). The morphology of nanoparticles was characterized by transmission electron microscopy (TEM) using negative staining. The formulation

stability was evaluated by monitoring the change of size distribution at 4 °C for one week.

## In vitro generation of TAMCs

Bone marrow progenitor cells were collected from tibias and femurs of C57BL/6 mice as described in[22]. Cells were seeded into 6 well plate with a density of $1.2 \times 10^6$ cells per well in complete RPMI [RPMI-1640 (Corning) containing 10% FBS, 1% penicillin and streptomycin, 1% HEPES (Sigma), 1% nonessential amino acids (Gibco), 1% sodium pyruvate (Coring), and 0.1% 2-mercaptoethanol (Gibco)] with the addition of M-CSF (PeproTech) at 40 ng/ul. After 3 days, media was replaced by 50% complete RPMI and 50% conditioned media (collected from CT-2A cell culture 72 h post original seeding with $2 \times 10^6$ CT-2A cells) and maintained for additional 3 days. Immunosuppressive phenotype of in vitro generated TAMCs was determined by flow cytometry and CD8[+] T cell suppression assay as previously described[46] (Supplementary Fig. 6).

## In vitro tethering, phagocytosis, and antigen presentation assay

For binding assay, CT-2A cells were cultured for 72 h following 9 Gy of irradiation using a RS 2000 Irradiator (Rad Source), labeled with Wheat Germ Agglutinin-Alexa Fluor 488 (Fisher), and co-cultured at 1:2 ratio with Wheat Germ Agglutinin-Alexa Fluor 647 (Fisher)-labeled TAMCs in glass bottom confocal dishes (World Precision). Cells were treated with Rhod-labeled B-LNP at an equivalent αCD47 concentration of 10 μg/ml for 0.5 h at 37 °C. After washed, the cells were observed under a Leica DMi8 microscope. Data was processed using ImageJ. For phagocytosis assay, CT-2A-GFP cells (+/− RT of 9 Gy 72 h prior to assay) were co-cultured with CellTracker Red CMTPX (Fisher)-labeled TAMCs in serum-free RPMI. αCD47 or B-LNP at an equivalent αCD47 concentration of 10 μg/ml was added to the co-culture and incubated for 4 h at 37 °C. Phagocytosis index was calculated by the ratio of TAMCs containing phagocytosed tumor cells/total number of TAMCs. Alternatively, cells were plated in an ultra-low attachment 24-well plate (Corning) for clustering assay. The kinetics of cell-cell interaction was quantified using an Incucyte S3 Live Cell Analysis System (Sartorius) over a time course of 17 h. Quantification was performed using Incucyte Cell-By-Cell Analysis Software. For antigen presentation, CT-2A-OVA cells were cultured for 72 h following 9 Gy of RT and co-cultured at 1:2 ratio with TAMCs. Cells were treated with αCD47 or B-LNP at an αCD47concentration of 10 μg/ml for 48 h. TAMC OVA-peptide SIINFEKL presentation by MHC class I (H-2Kb) was measured by flow cytometry.

## Orthotopic mouse models of GBM, cannula implantation, and radiotherapy

Six to eight weeks old mixed-gender mice were implanted with CT-2A ($5 \times 10^4$ cells per mouse, unless otherwise specified) or PVPF8 ($7.5 \times 10^4$ cells per mouse) glioma cells using a stereotactic apparatus following the exact specifications as described previously[22,50]. For multiple intracranial administration of therapeutics and/or nanoparticles, cannulas were implanted following the procedures as described in[22]. For brain-focused radiotherapy, mice were exposed to a 3 Gy daily dose of irradiation, using a Gammacell 40 Exactor (Best Theratronics), for three consecutive days starting on the seventh day after tumor cell implantation. Standard post-surgery care was given following the IACUC-approved protocol. The animals were euthanized via $CO_2$ asphyxiation followed by cervical dislocation once they reached the endpoints, as defined by the presence of severe neurological deficits due to the tumor burden. These deficits include inabilities to maintain an upright position, spasticity, seizures, circling, paralysis, paresis or an inability to ambulate such that access to food and/or water is prevented. Other early termination endpoint criteria include those stated in the IACUC approved body condition scoring system (body condition score <2).

## Isolation of glioma-infiltrating immune cells

Brain tumor-bearing mice were euthanized and perfused intracardially with 5 ml of cold DPBS. Single cell suspension of brain/tumor was processed using a tissue homogenizer (Potter-Elvehjem PTFE pestle) in Hanks' balanced salt solution (HBSS, Gibco), followed by 30/70 Percoll gradient separation to remove myelin and debris (GE Healthcare). Immune cells were collected into complete RPMI for immune phenotyping or ex vivo assays.

## In vitro and ex vivo TAMC targeting

In vitro generated TAMCs were seeded in 6-well plates and incubated with Rhod-labeled B-LNP (0.1 mg lipids/ml) at 4 °C. At predetermined time intervals, the cells were gently washed with ice-cold PBS thrice, stained with viability dye, and analyzed by flow cytometry. For ex vivo targeting, single cell suspension collected from the tumor-bearing brains was plated into 96 well U bottom plates, followed by incubation with Rhod-labeled B-LNP for 1 h at 4 °C. All cells were blocked with anti-CD16/32 antibodies (BioLegend) before incubation. To reveal the targeting mechanism, selected groups of cells were also pre-incubated with an excess amount of anti-PD-L1 antibody.

## In vitro analysis of diABZI effects on TAMC and T cell

In vitro generated TAMCs ($1 \times 10^6$ cells/well) in 6-well plate were treated with diABZI at 200 nM. After 6 h of treatment, cells were washed and collected for RNA isolation using the RNeasy Plus Mini Kit (Qiagen). Total RNA was quantified by Nanodrop (Thermo Scientific), and cDNA was synthesized using the iScript cDNA synthesis kit (Bio-Rad). Gene expression was analyzed by quantitative PCR analysis (Bio-Rad). The primer sequences are listed in Supplementary Table 1. Alternatively, flow cytometric analysis was performed 24 h after treatment. To determine the cytotoxicity of diABZI to T cells, T cells were isolated using mouse T cell isolation kit (STEMCELL Technologies) and seeded at $2 \times 10^5$ cells/well in 96-well U bottom plates with the addition of Dynabeads mouse T cell activator CD3/CD28 (Fisher) and IL-2 (PeproTech) at 50 U/ml. Cells were treated with diABZI at 200 nM for 48 h and collected for annexin V assay (BioLegend) using flow cytometry.

## In vivo CD8[+] T cell transfer

C57BL/6 mice were intracranially implanted with $5 \times 10^4$ CT-2A cells and received a whole-body irradiation of 9 Gy for lymphopenia seven days post-tumor implantation. Certain groups of mice also received B-LNP/diABZI treatment at 0.25 mg diABZI/kg through cannulas 2 days post-RT. One day after last nanoparticle treatment, all mice received intravenously administrated $5 \times 10^6$ CD45.1[+] CD8[+] T cells isolated from CT-2A-bearing CD45.1[+] mice. Three days after adoptive cell transfer, CD8[+] T cells in tumor-bearing brains were evaluated by flow cytometry using the anti-mouse CD45.1 PE-Cy7 and total CD45.2 APC from BioLegend. Activation of T cells was evaluated by anti-CD69 Pacific Blue.

## In vivo B-LNP/diABZI treatment combined with radiotherapy

C57BL/6 mice with intracranially implanted CT-2A ($5 \times 10^4$ cells per mouse) were randomly grouped and treated with saline, B-LNP, B-LNP/diABZI, or a combination of free diABZI + αPD-L1 + αCD47 (0.25 mg diABZI per kg) on the ninth- and thirteenth-day after tumor implantation through intracranially implanted cannula. Certain groups of mice also received brain-focused radiotherapy as described above. Supportive care of mice post-tumor implantation and treatments was provided in full compliance with the approved animal protocols. Mice were euthanized for ex vivo studies or followed for their survival following outlined endpoint protocols. Long-term survivor (LTS) mice were euthanized 100 days after initial tumor implantation. Age-matched tumor-bearing mice were euthanized 14 days after CT-2A tumor implantation. Brains were collected for immunophenotypic analysis by flow cytometry. Tumor burden was analyzed by

hematoxylin and eosin (H&E) staining. Therapeutic effects in STING^Gt mice were evaluated following the same procedure. Rag1^-/- mice were also used to determine the role of adaptive immunity in the treatment outcomes following the same procedure, except that one day after each B-LNP/diABZI treatment, selected groups of mice also received intravenous administration of 4×10^6 CD8^+ T cells isolated from CT-2A-bearing C57 mice using the mouse CD8^+ T cell isolation kit (STEMCELL Technologies).

## Murine immunophenotypic analysis

All antibodies were purchased from BioLegend and used at a 1:200 dilution except as otherwise specified. Single cell suspensions were pre-blocked with anti-CD16/32 antibodies before staining with antibody panels as described below. Dead cells were excluded using the Fixable Viability Dye eFluor780 (Fisher). CD45 BV510 and CD11b BV711 were used to determine the myeloid and lymphocyte compartments. The Foxp3 fixation/permeabilization (Invitrogen) protocol was used for intracellular staining. For cytokine staining, cells were pre-incubated with cell stimulation cocktail plus protein transport inhibitors (Fisher) for 4 h. All acquisition was performed using a BD FAC-Symphony flow cytometer. Flow cytometry data were collected using FACS DIVA software and analyzed by FlowJo software. Exemplifying flow cytometry gating strategy was demonstrated in Supplementary Fig. 21. Unless otherwise stated in the axis labels, the following panels were used. For myeloid compartment analysis: anti-CD45 BV510, anti-CD11b BV711, anti-CD80 BV605, anti-CD86 Alexa Fluor 700, anti-CD40 Pacific Blue, anti-CD206 FITC, anti-ARG1 APC (Fisher), anti-iNOS PE, CD11c PerCP-Cy5.5. For lymphocytic compartment analysis: anti-CD45 BV510, anti-CD11b BV711, anti-CD8 BV605, anti-CD4 PE-Cy7, anti-CD44 FITC, anti-CD62L PerCP-Cy5.5, anti-CD69 Pacific Blue, anti-CD25 Alexa Fluor 700, anti-PD-1 PE. For LTS analysis: anti-CD45 BV510, anti-CD11b BV711, anti-CD8 BV605, anti-CD4 PE-Cy7, anti-CD44 FITC, anti-CD62L PerCP-Cy5.5, anti-PD-1 PE, anti-GzmB APC, anti-IFN-γ Alexa Fluor 700. For tumor cell analysis: anti-CD47 PE, anti-CRT Alexa Fluor 488 (Cell Signaling, 1:50), anti-PD-L1 APC. Detailed antibody information is provided in Supplementary Table 2.

## Meso Scale Discovery (MSD) cytokine analysis

Fifteen days after CT-2A tumor implantation, blood and brain tumors were collected from the CT-2A-bearing mice received RT +/- B-LNP/diABZI treatments as the treatment regimen described above. Protein was extracted from brain tumors following a previously published protocol[51]. Cytokines were measured using a V-PLEX Cytokine Panel 1 Mouse Kit following the manufacturer protocol (Meso Scale Diagnostics).

## Single-cell RNA sequencing analysis

Sixteen days after CT-2A tumor implantation, tumors were collected from the brains of CT-2A-bearing mice received RT +/- B-LNP/diABZI treatments as the treatment regimen described above. 5 mice were pooled for each n reported. Single cell suspension was processed using the adult brain dissociation kit and gentleMACS dissociator (Miltenyi Biotec), followed by Fc-blockade using anti-CD16/32 antibodies (Bio-Legend) and anti-CD45 magnetic bead-based separation (Miltenyi Biotec). Tumor-infiltrating immune cells were enriched by mixing CD45^+ and CD45^- cells at a 9:1 ratio. Barcoding, library preparation, and sequencing were generated and analyzed by the NUSeq Core at Northwestern University. To visualize the results, the normalized gene barcode matrix was used to compute a neighborhood graph of cells, then Uniform Manifold Approximation and Projection (UMAP) was performed with default parameters. The whole pipeline was implemented using Scanpy[52]. Cell type annotation was performed using mouse RNA-seq reference and R package SingleR[53]. Differential expression analysis was performed by Wilcoxon rank-sum test between treatment groups. R package AUCell was used to calculate enrichment score for each of the gene sets in single-cell RNA-seq data[54]. AUCell calculates "Area Under the Curve" (AUC) to assess the enrichment of the input gene set within the expressed genes for each cell using ranking based score method. Gene sets were obtained from MSigDB v7.1, utilizing the C5: GO gene sets collection[55].

## Metabolite isolation and LC-MS/MS analysis

Single cell suspensions from tumor-bearing brains were prepared using an adult brain dissociation kit and gentleMACS dissociator (Miltenyi Biotec), followed by Fc-blockade using anti-CD16/32 antibodies (BioLegend) and anti-Gr1 magnetic bead-based isolation (Miltenyi Biotec) as previously described[46]. Cell isolates were washed with PBS, and metabolites from 1×10^6 Gr1^+ cells were extracted with 80% methanol/20% H_2O. Samples were dried via Speedvac (Fisher) followed by resuspension in 50% acetonitrile.

Samples were analyzed by high-performance liquid chromatography and high-resolution mass spectrometry and tandem mass spectrometry (HPLC-MS/MS). Specifically, a Thermo Q-Exactive in line with an electrospray source Ultimate3000 series HPLC (Fisher) consisting of a binary pump, degasser, and auto-sampler outfitted with an Xbridge Amide column (Waters; dimensions of 4.6 mm × 100 mm and a 3.5 μm particle size). The mobile phase A contained 95% water, 5% acetonitrile in water, 20 mM ammonium hydroxide, 20 mM ammonium acetate, pH = 9.0; B was 100% acetonitrile. The gradient was as following: 0 min, 15% A; 2.5 min, 30% A; 7 min, 43% A; 16 min, 62% A; 16.1-18 min, 75% A; 18-25 min, 15% A with a flow rate of 400 μl/min. The capillary of the ESI source was set to 275 °C, with sheath gas at 45 arbitrary units, auxiliary gas at 5 arbitrary units and the spray voltage at 4.0 kV. In positive/negative polarity switching mode, an m/z scan range from 70 to 850 was chosen and MS1 data was collected at a resolution of 70,000. The automatic gain control (AGC) target was set at 1×10^6 and the maximum injection time was 200 ms. The top 5 precursor ions were subsequently fragmented, in a data-dependent manner, using the higher energy collisional dissociation (HCD) cell set to 30% normalized collision energy in MS2 at a resolution power of 17,500. Data acquisition and analysis were carried out by Xcalibur 4.1 and Tracefinder 4.1 software, respectively (Thermo Fisher Scientific). For differential metabolite analysis, additional peak-area normalization was performed by total ion count to verify data integrity. After normalization, data was imputed into Metaboanalyst software (https://www.metaboanalyst.ca/). Heat maps were generated by one-factor statistical analysis using normalized peak-areas data generated via LC-MS/MS. The top 100 metabolites were sorted by p-value for visualization. For pathway analysis, all metabolite identities were converted to KEGG identifiers and pathway impact was automatically generated from the same data as above.

## CAR T cell brain tumor migration

A replication-deficient retrovirus was generated by transfecting Phoenix-ECO cells with a previously developed IL13Ra2.CAR plasmid according to our established protocol[56]. CD3^+ T cells were isolated from the spleens of 6 to 8 weeks old C57BL/6 mice using a mouse T cell isolation kit (STEMCELL Technologies). Activated T cells were transduced with retroviral supernatants for two consecutive days. In order to generate a CT-2A cell line expressing IL13Ra2 (CT-2A-IL13Ra2), CT-2A cells were transduced with pEF6/myc-His vector encoding human IL13Rα2[57] according to previously developed protocol[56]. For in vivo experiment, 6 to 8 weeks old Rag1^-/- mice were injected with 2×10^5 CT-2A-IL13Ra2 cells. Fourteen days later, mice were treated with 3 daily doses of irradiation at 3 Gy each and followed by B-LNP/diABZI treatment (0.25 mg diABZI per kg) through cannulas 24 h post-last dose of radiation. 24 hours after nanoparticle treatment, IL13Ra2.CAR T cells were intravenously administrated at 1×10^7 cells per mouse. Seventy-two hours later, mice were euthanized and perfused with cold PBS. The brains were extracted

and fixed in 10% formalin. Paraffin tissue sections (4 μm) of the brains were stained with H&E for histological evaluation of tissue and with anti-CD3 antibody (Abcam ab16669; 1:1000) for detection of CAR T cells migrated to the brain.

### In vivo B-LNP/diABZI combined with Stupp treatment
C57BL/6 mice with intracranially implanted CT-2A (5×10⁴ cells per mouse) were randomly grouped and received a daily dose of brain-focused irradiation at 3 Gy and TMZ at 50 mg per kg intraperitoneally for three consecutive days starting on the seventh day after tumor cell implantation. Certain groups of mice also received B-LNP/diABZI (0.25 mg diABZI per kg) or a combination of free diABZI and therapeutic antibodies (αPD-L1 + αCD47) on the ninth and thirteenth day after tumor implantation through intracranially implanted cannula. All mice were followed for their survival following outlined endpoint protocols.

### In vivo nanoparticle treatment through systemic delivery
C57BL/6 mice with intracranially implanted CT-2A (7.5×10⁴ cells per mouse) were randomly grouped and treated with saline or P-LNP/diABZI (2.5 mg diABZI per kg) through intravenous administration on the ninth, fourteenth, and nineteenth day after tumor implantation. Selected groups of mice also received brain-focused radiotherapy for three consecutive days starting on the sixth day post-tumor implantation. Supportive care of mice post-tumor implantation and treatments was provided in full compliance with approved animal protocols. Mice were euthanized for ex vivo studies or followed for their survival following outlined endpoint protocols. 180 d after initial tumor implantation, surviving mice were rechallenged with 7.5×10⁴ CT-2A cells in the opposite hemisphere (left) to the initial injection site. LTS mice were euthanized 100 days after initial tumor implantation. Age-matched tumor-bearing mice were euthanized 14 days after CT-2A tumor implantation. Brains were collected for immunophenotypic analysis by flow cytometry. Tumor burden was analyzed by H&E staining. Therapeutic effects of P-LNP/diABZI was similarly evaluated in mice implanted with PVPF8 cells (7.5×10⁴ cells per mouse) and received RT and intravenous administrations of saline or P-LNP/diABZI (2.5 mg diABZI per kg) on the eighth and thirteenth day after tumor implantation.

### Patient-derived TAMC and autologous CD8⁺ T cell activation
Two freshly resected tumor specimens from GBM patients were involved in this study. Temozolomide and radiotherapy were received post resection. Characteristics of GBM patient samples are provided in Supplementary Table 3. Tumor specimens were diced using a razor blade. Single-cell suspensions were processed using a brain tumor dissociation kit and gentleMACS dissociator (Miltenyi Biotec). Single cell suspension was pre-blocked with Human TruStain FcX (BioLegend) for 15 min at 4 °C and labeled with biotinylated anti-CD45 antibodies (BioLegend). The cells were washed and then incubated with anti-biotin magnetic beads (Miltenyi Biotec), followed by positive selection using MS columns (Miltenyi Biotec). Peripheral blood was collected in EDTA-treated tubes, and patient PBMCs were isolated by Ficoll gradient (GE Healthcare). T cells were isolated from PBMCs using the Human T cell isolation Kit (STEMCELL Technologies). 7.5×10⁵ of CD45⁺ tumor-infiltrating immune cells and 2.5×10⁵ of CD45⁻ tumor cells were mixed in 24-well plate. 2.5×10⁵ of peripheral T cells labeled with eFluor 450 dye were added into a cell culture insert with pore size of 3.0 μm (Sigma). B-LNP/diABZI was added into the bottom chamber at a diABZI concentration of 100 nM. Cells were incubated for 72 h. The following flow cytometry panel was used for human patient sample analysis: anti-CD45 BV510, anti-CD11b PE, anti-CD3 PE-Cy7, anti-CD8 Alexa Fluor 700, anti-CD4 FITC, anti-CD69 PerCP-Cy5.5, anti-CD25 APC, anti-CD86 BV711, anti-PD-1 BV605. All antibodies were purchased from Biolegend and used at 1:50 dilution. Cells were pre-blocked with

Human TruStain FcX for 15 min at 4 °C before antibody staining. Dead cells were excluded from the analysis using the eBioscience Fixable viability dye eFluor780. All acquisition was performed using a BD FACSymphony flow cytometer and analyzed using FlowJo software.

### Statistical analysis
All statistical analyses were performed with Prism Graph-Pad 9.5 Software. Two-sided Student's t test was used for comparing the two groups. Multiple groups were analyzed by one-way ANOVA, followed by Tukey's post hoc test. Longitudinal data from multiple groups were analyzed with two-way ANOVA, followed by Tukey's post hoc test. All numerical data were reported as mean +/- SEM. Kaplan-Meier plots were generated to determine relative survival of tumor-bearing animals under different courses of treatments. p values for curve comparisons were calculated using Log-rank test for groups with proportional hazards or Renyi's test for groups with crossing hazards, followed by Bonferroni correction.

### Reporting summary
Further information on research design is available in the Nature Portfolio Reporting Summary linked to this article.

### Data availability
The single-cell RNA sequencing data files have been deposited in the NCBI Sequence Read Archive (SRA) database under the accession number PRJNA903231. The metabolomics data have been deposited in the Metabolomics Workbench under the Project ID PR001593 (https://doi.org/10.21228/M8SH9M). Source data are provided with this paper.

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

## Acknowledgements

We would like to thank the Northwestern Nervous System Tumor Bank for providing samples from patients with glioblastoma. We would also like to thank Northwestern University Flow Cytometry Core Facility, Analytical bioNanoTechnology Core Facility, NUSeq Core Facility, Immunotherapy

Assessment Core Facility, Metabolomics Core Facility, and Mouse Histology and Phenotyping Laboratory. This work was supported by National Institutes of Health (NIH)/National Cancer Institute (NCI) grant (R01CA266487) and Northwestern Brain Tumor SPORE Career Enhancement Program (P50CA221747) to P.Z., NCI Outstanding Investigator Award (R35CA197725) to M.S.L., NIH grants to M.S.L (P50CA221747), to M.S.L and J.M (R01NS115955), and to A.M.S. (R01NS110703).

## Author contributions

P.Z., J.M., and M.S.L. conceived the study; P.Z., J.M., C.L.-C., A.R., C.S., H.W., B.C., A.E., M.Z., C.D., R.L., and A.C. performed and analyzed experiments. J.Z. and T.X. provided assistance with bioinformatics and statistical analysis. Y.H. provided assistance with animal surgeries and reagent preparations. A.L.-R. performed all animal breeding for the study. A.M.S. and I.V.B contributed new reagents/analytic tools and contributed to manuscript preparation. P.Z., J.M., and M.S.L. provided critical feedback, contributed to manuscript preparation, and oversaw the research program. All authors listed reviewed the manuscript and provided feedback with writing and revisions.

## Competing interests

A provisional patent application pertaining to the work presented in this manuscript was filed by Northwestern University with P.Z. and M.L. as inventors. The remaining authors declare no competing interests.
