## [Peer Review File · Nature Communications]

STING agonist-loaded, CD47/PD-L1-targeting nanoparticles
potentiate antitumor immunity and radiotherapy for
glioblastomaREVIEWER COMMENTS

Reviewer #1 (expertise in nanoparticles in glioma, glioma therapy):

This manuscript by Zhang et al reports a nano-technology based platform to hijack myeloid cells and enhance the therapeutic efficacy of radiotherapy and immunotherapy for glioblastoma. The data is novel and rigorous and the experimental designs are adequate to test the hypothesis put forward by the authors. This manuscript will be of interest to the readership of Nature Communications. Please find enclosed below minor comments.

1- The graphical abstract could benefit of editing for clarity purposes, the font is too small and the mechanistic details need to be further clarified.

2- Immunofluorescence images shown in Fig 2, panels h and I could benefit from more in depth analysis using Z-stacking to conclusively demonstrate phagocytosis.

3- Scale bars need to be clearly indicated in Fig 2, J they are almost not visible.

4- If additional samples are available, it would be very interesting to depict markers of exhausting in the tumor infiltrating T cells in response to the treatment. If not available, please include a comment in the discussion.

5- The histological images shown in Figure 6 could be improved. Also, higher magnification images could be included and scale bars need to be added to all panels.

6- If the authors have collected serum or plasma from the treated animals, it would be interesting to include the classical cytokine panel to demonstrate immunogenic cell death induced by their treatment. Otherwise, please include in the discussion.

Reviewer #2 (expertise in CD47, macrophage signalling):

Comments for Author

The authors generated a bispecific-lipid nanoparticle (B-LNP) that was functionalized with anti-CD47/PD-L1 dual-targeting capability. In B-LNP, anti-CD47 antibody was utilized for targeting and blocking of overexpressed anti-phagocytotic molecule CD47 in irradiated glioma cells. Anti-PD-L1 antibody was utilized for binding and engaging of tumor-associated myeloid cells (TAMCs) to glioma cells. They found that B-LNP promoted phagocytotic activity of TAMCs against irradiated glioma cells. They next generated diABZI (an agonist for stimulator of interferon genes) loaded B-LNP (B-LNP/diABZI) to boost subsequent T cell recruitment and antitumor activity after engulfment of irradiated glioma cells by TAMCs. They demonstrated that B-LNP/diABZI remodeled TAMC functionality from immunosuppressive to immunostimulatory. B-LNP/diABZI-reprogrammed TAMCs enhanced infiltration and activation of CD8-positive T cells in glioblastoma. Administration of B-LNP/diABZI improved the antitumor effects of radiotherapy and elicited long-term immune responses against glioma in glioblastoma-bearing mice. Finally, they tested the possibility of administering LNP with anti-PD-L1 antibody (P-PLNP)/diABZI through a systemic delivery approach.

Overall, their experiments are well performed, and their findings would contribute to more effective strategies to treat glioblastoma. This reviewer has some concerns, however.

Major comments:

1. The CD47-SIRP α interaction is known to suppress the phagocytotic activity of phagocytes. Does B-LNP block the CD47-SIRP α interaction? The authors should show the significance of blockade of the CD47-SIRP α interaction by B-LNP in terms of its anti-tumorous effects on glioblastomas.
2. In Fig 6, how does intravenously injected P-LNP/diABZI reach TAMCs? Can P-LNP/diABZI cross the blood-brain barrier? The authors should demonstrate that intravenously injected P-LNP/diABZI reaches TAMCs in the mouse GBM tissues.

Minor comments:

1. In Fig. 1, CRT on the tumor cells looks like interacting with SIRP α on phagocytes. Is this true or previously shown? Otherwise, please correct the figure.
2. The authors should define abbreviations upon first appearance in the text. For example, "RT-elicited process" (page 2, line 26th-27th) should be "radiation therapy (RT)-elicited process".
3. In the right panel of Fig. 4h, it is difficult to find the dotted line which indicates border of normal brain and tumor site. Please draw the line thicker.

Reviewer #3 (expertise in CD47, glioblastoma):

I had the pleasure to review the article by Peng Zhang and colleagues on a novel nanotherapeutic treatment modality against glioblastoma.

Specifically, the group designed a bispecific nanoparticle that bridges GBM cells with myeloid cells via the don't eat me signal CD47 (tumor specificity, upregulated after RT) and PDL1 (myeloid specificity). Additional encapsulation of the nanoparticle with STING agonist diABZI, thereby providing myeloid specificity, boosted T cell recruitment and anti-tumoral efficacy, and synergized with radiotherapy.

The paper is very well written and uses state-of-the art concepts and methodologies to confirm their hypothesis. This includes a thorough in vitro and biophysical characterisation of the nanoparticle used, convincing in vivo animal work in conjunction with immunological/metabolic/transcriptional readouts, and potential translational applications.

Some modifications might improve the manuscript further:

Fig 2j: please comment on the low phagocytic index induced by aCD47 treatment alone. aCD47 treatment alone in CT2A-TAM co-cultures should normally already at least induce phagocytosis above baseline. Describe in the caption what modality was used for the assay (I guess incucyte). Maybe highlight phagocytic TAMCs in the micrographs since they are not readily obvious from what is depicted in Fig. 2j right panel.

Fig 2k: kinetic analysis looks very convincing, although I am also here surprised that aCD47 mediated CT2A phagocytosis without RT is basically absent. Here, B-LNP induced phagocytosis w/o RT is also not present, in contradiction to Figure 2j, where a significant phagocytic index is displayed. Any explanations for these discrepancies? Furthermore, statistical information in the plot are missing, or at least it would be good to have error bars in the area measurements, since I guess multiple fields of view were evaluated.

Fig 3a: the binding of B-LNP to microglia (and also the other cell types) seems very significant. This is not highlighted in the plot.

Fig 3c: As far as I understand, the RNAseq analysis was performed with 90% CD45+ and 10% CD45- cells. It is interesting to see that in the Combo condition, practically no CD45- tumor cells are visible in the composition plot. Please provide an explanation.

Furthermore, microglia, which are the prominent brain intrinsic phagocytic population, seem to be underrepresented in these CT2A tumors, which is a little bit in contrast to our own data. I understand that this is a myeloid rich tumor, but microglia can still not be neglected here. I am worried that the amount of microglia might be higher, especially when also adjacent brain parenchyma was sampled. Did you also perform a microglia-centered GO analysis? Are microglia, in general, also affected positively by B-LNP treatment?

A bigger concern in this experiment for me is the lack of a B-LNP without diABZI loading (empty B-LNP) control. I understand that this would be a major additional experimental effort, but this absence has to be discussed as a limitation at least, or a clear statement on why this data is not provided here has to be made.

Fig 4: the T cell analysis is convincing with the caveat that the DiABZI-free B-LNP condition is

missing as well. In the caption of Fig 4a it should say 'B-LNP/DdiABZI programmed'...

Fig 4i: a treatment scheme would help the reader (this was local application of antibody cocktails, I assume, should be highlighted in the figure).

Fig 4j: it should say 'tumor-infiltrating'. I would update the scheme by adding the therapeutic modality applied.

Fig 5a: the condition B-LNP without diABZI is missing, it would be great to include if possible, as discussed above, or discuss as a limitation.

Fig 6/systemic treatments: immunological effects of B-LNP/diABZI are clearly visible. Can the authors demonstrate homing of the nanoparticles to the tumor, by e.g. showing pRhodo Stained B-LNPs after iv injections in CT2A tumors in an additional figure? Any longitudinal/pharmacokinetic data would enhance the ms.

CAR T cell data and enhanced T-cell infiltration into the tumor after B-LNP/diABZI treatment: while the observed effect of enhanced T-cell homing is very interesting, the authors do not provide direct mechanistic clues on how this is working. Local proinflammatory cytokines certainly play a role, but the homing has to be induced in the periphery. Is there a chance to look at peripheral cytokines/chemokines that might explain the effect?

Why was the CART cell experiment performed in RAG1-/- mice and not in wildtype mice - please provide a short explanatory sentence.

Discussion: The term 'B-LNP/diABZI' should be used consistently, eg on line 332, 337 etc.

Translational potential: I would not only focus on iv delivery (and, as mentioned above, tumor homing of B-LNP needs to be demonstrated). The effect seems higher when the drug is administered locally, which is also clinically feasible, and can be discussed as an adjuvant to radiation therapy or in case of recurrent GBM. Limitations as mentioned above need to be highlighted.

Minor comments:

methods line 511: please provide the cell amount of CT2A and PF8 cells injected per mouse.

Overall, the work by Peng et al is very interesting and merits publication in Nature Communications after addressing the above mentioned concerns.

RESPONSE TO REVIEWERS' COMMENTS

Dear Reviewers,

Thank you very much for spending time reviewing our manuscript and for the invaluable comments. We have attempted to address all the questions. Following the suggestions, we have involved critical controls, e.g., drug-free B-LNPs, to evaluate the antitumor effects and the impact on the tumor immune microenvironment. We have also assessed nanoparticles through the systemic delivery route in terms of homing to brain tumors and targeting TAMCs by live animal imaging and immunofluorescence analysis. Thanks to your expert insights, we have demonstrated that our treatments increased a panel of pro-inflammatory cytokines in both serum and the brain tumor milieu, particularly T cell-recruiting/activating cytokines, providing a mechanistic clue of the nanoparticle-stimulated brain tumor infiltration of T cells, including CAR T cells. Following the advice, we have also re-evaluated our immunofluorescence, scRNA-seq, and flow cytometry analyses, allowing an in-depth understanding of the effects of our treatments on the glioma immune microenvironment, e.g. microglia and T cell exhaustion. The changes have been marked in red in the revised manuscript. A point-by-point response to the comments of the reviewers was listed as detailed below:

Response to comments of Reviewer #1:

Q1: The graphical abstract could benefit of editing for clarity purposes, the font it too small and the mechanistic details need to the further clarified.

A1: We would like to thank the reviewer for the suggestion. We have improved the graphical abstract (**Fig. 1**) accordingly in the revised manuscript.

Q2: Immunofluorescence images shown in Fig 2, panels h and I could benefit from more in depth analysis using Z-stacking to conclusively demonstrate phagocytosis.

A2: We would like to thank the reviewer for the expert insights. Following the advice, we have included Z-stack imaging in both co-culture assays. We agree with the reviewer that Z-stack imaging indeed provides an in-depth analysis to better demonstrate that B-LNP serves as a “bridge” between glioma cells and TAMCs to promote the formation of cell-cell tethering. We have included these data in the revised manuscript as **Fig. 2i** and **Supplementary Fig. 3**.

Q3: Scale bars need to be clearly indicated in Fig 2, J they are almost not visible.

A3: We have changed the scale bars in **Fig. 2j** accordingly.

Q4: If additional samples are available, it would be very interesting to depict makers of exhausting in the tumor infiltrating T cells in response to the treatment. If not available, please include a comment in the discussion.

A4: We would like to thank the reviewer for the expert insights. Flow cytometric analysis of glioma-bearing brains shows an increased percentage of PD-1⁺ LAG3⁺ CD8⁺ T cells after B-LNP/diABZI or cocktail treatments. These data may indicate a potential benefit of combining immune checkpoint inhibitors with our therapy to reverse the post-therapy T cell exhaustion. We have included these results in the revised manuscript as **Supplementary Fig. 10**.

Q5: The histological images shown in Figure 6 could be improved. Also, higher magnification images could be included and scale bars need to be added to all panels.

A5: We have provided high magnification images with scale bars in **Fig. 6** in the revised manuscript accordingly.

Q6: If the authors have collected serum or plasma from the treated animals, it would be interesting to include the classical cytokine panel to demonstrate immunogenic cell death induced by their treatment, Otherwise, please include in the discussion.

A6: We would like to thank the reviewer for the expert insights. We have assessed the classical cytokine panel in serum and brain tumor samples using an MSD V-PLEX Cytokine Panel 1 Mouse Kit, including CXCL10, CCL2, CCL3, CXCL2, IL-27p28/IL-30, IL33, IL15, IL9, and IL17A/F. Our results indicate that the B-LNP/diABZI treatment increased a panel of pro-inflammatory cytokines in both serum and the brain tumor milieu, particularly T cell-recruiting/activation cytokines. We have included these data in the revised manuscript as **Fig. 4i** and **Supplementary Fig. 9**.

Response to comments of Reviewer #2:

Q1: The CD47-SIRP α interaction is known to suppress the phagocytotic activity of phagocytes. Does B-LNP block the CD47-SIRP α interaction? The authors should show the significance of blockade of the CD47-SIRP α interaction by B-LNP in terms of its anti-tumorous effects on glioblastomas.

A1: We would like to thank the reviewer for the expert insights. Our results indicate that B-LNP blocks the CD47 signals in glioma cells (**Fig. 2e**), inducing promoted phagocytic clearance of glioma cells (**Fig. 2k**). Following the reviewer's suggestion, we also evaluated the antitumor effects of B-LNP in glioma-bearing mice (**Supplementary Fig. 14**). Our data indicate a moderately prolonged survival of CT-2A glioma-bearing mice receiving B-LNP treatments (median survival: w/o B-LNP, 29 days; B-LNP, 38 days). These data are consistent with our results using STING^{Gt} mice and RAG^{-/-} mice in which the STING-related effects are absent (**Fig. 5b, c**).

Q2: In Fig 6, how does intravenously injected P-LNP/diABZI reach TAMCs? Can P-LNP/diABZI cross the blood-brain barrier? The authors should demonstrate that intravenously injected P-LNP/diABZI reaches TAMCs in the mouse GBM tissues.

A2: We would like to thank the reviewer for the expert insights. We have accordingly assessed the ability of nanoparticles to cross the blood-brain barrier for brain distribution by live animal

imaging (**Supplementary Fig. 16a, b**). Our results indicate that the intravenously injected nanoparticles were rapidly distributed to brain tumor sites, likely due to the enhanced permeability and retention effect (passive targeting) (PMID: 22595146) because of the small size of nanoparticles and due to the TAMC-targeting capability (active targeting) because of the ligand functionalization. The TAMC targeting was further demonstrated by the co-localization of nanoparticles and TAMCs in brain tumors as evaluated by immunofluorescence staining of tumor-bearing brains (**Supplementary Fig. 16c**). These data altogether indicate that intravenously injected nanoparticles could reach TAMCs in mouse glioma tissues.

Q3: In Fig. 1, CRT on the tumor cells looks like interacting with SIRP α on phagocytes. Is this true or previously shown? Otherwise, please correct the figure.

A3: We apologize for the mislabeled scheme and have revised the figure accordingly.

Q4: The authors should define abbreviations upon first appearance in the text. For example, “RT-elicited process” (page 2, line 26th-27th) should be “radiation therapy (RT)-elicited process”.

A4: We would like to thank the reviewer for the comment and have defined all the abbreviations in the revised manuscript accordingly.

Q5: In the right panel of Fig. 4h, it is difficult to find the dotted line which indicates border of normal brain and tumor site. Please draw the line thicker.

A5: We would like to thank the reviewer for the comment and have improved the figure accordingly.

Response to comments of Reviewer #3:

Q1: Fig 2j: please comment on the low phagocytic index induced by α CD47 treatment alone. α CD47 treatment alone in CT2A-TAM co-cultures should normally already at least induce phagocytosis above baseline. Describe in the caption what modality was used for the assay (I guess incubate). Maybe highlight phagocytic TAMCs in the micrographs since they are not readily obvious from what is depicted in Fig. 2j right panel.

A1: We would like to thank the reviewer for the expert insights. One interpretation of the low phagocytic index caused by α CD47 treatment alone is the low calreticulin expression in murine glioma cells without RT (**Fig. 2b**). It has been shown that efficient phagocytosis of tumor cells requires blockade of anti-phagocytic signal CD47 as well as expression of pro-phagocytic molecule calreticulin (PMID: 21178137). Following the reviewer’s advice, we have highlighted the phagocytic TAMCs in the micrographs and revised the caption accordingly.

Q2: Fig 2k: kinetic analysis looks very convincing, although I am also here surprised that α CD47 mediated CT2A phagocytosis without RT is basically absent. Here, B-LNP induced phagocytosis w/o RT is also not present, in contradiction to Figure 2j, where a significant phagocytic index is displayed. Any explanations for these discrepancies? Furthermore, statistical information in the

plot are missing, or at least it would be good to have error bars in the area measurements, since I guess multiple fields of view were evaluated.

A2: We would like to thank the reviewer for the expert insights. We agree with the reviewer that the effect of α CD47 and B-LNP on inducing phagocytosis without RT is not apparent, which might be due to the low expression of calreticulin in murine glioma cells without irradiation. There was indeed a difference in readouts between **Fig. 2j** and **Fig. 2k**. A potential explanation comes from the different methods used in these assays. Following a published protocol (PMID: 21178137), the ratio of phagocytic cells was used to determine the phagocytosis index in **Fig. 2j**. As an additional method, the change of glioma cell surface area over time was monitored to determine the phagocytic clearance of glioma cells in **Fig. 2k**. We also apologize for the missed statistical information and have updated the graph accordingly.

Q3: Fig 3a: the binding of B-LNP to microglia (and also the other cell types) seems very significant. This is not highlighted in the plot.

A3: We would like to thank the reviewer for the expert insights. We agree with the reviewer that microglia, an important component of the myeloid compartment within brain tumors, also efficiently take up nanoparticles. We have highlighted the B-LNP binding to microglia in the plot (**Fig. 3a**) and discussed the critical role of microglia in our therapy in the revised manuscript accordingly.

Q4: Fig 3c: As far as I understand, the RNAseq analysis was performed with 90% CD45+ and 10% CD45- cells. It is interesting to see that in the Combo condition, practically no CD45- tumor cells are visible in the composition plot. Please provide an explanation. Furthermore, microglia, which are the prominent brain intrinsic phagocytic population, seem to be underrepresented in these CT2A tumors, which is a little bit in contrast to our own data. I understand that this is a myeloid rich tumor, but microglia can still not be neglected here. I am worried that the amount of microglia might be higher, especially when also adjacent brain parenchyma was sampled. Did you also perform a microglia-centered GO analysis? Are microglia, in general, also affected positively by B-LNP treatment? A bigger concern in this experiment for me is the lack of a B-LNP without diABZI loading (empty B-LNP) control. I understand that this would be a major additional experimental effort, but this absence has to be discussed as a limitation at least, or a clear statement on why this data is not provided here has to be made.

A4: We fully agree with the reviewer that we observed very few tumor cells in the Combo group. The tumors were very small or barely visible when we dissected the tumor tissues from brains for scRNA-seq, while the RT groups had quite big tumors. This reduced tumor size was related to the antitumor effect of the Combo treatment in vivo, which was also reflected by the prolonged animal survival and eradication of gliomas (**Fig. 5**).

We would like to thank the reviewer for the expert insights into microglia. We agree with the reviewer that microglia are a prominent brain intrinsic phagocytic population (PMID: 30602457) and they may have been underrepresented in our experiment. A potential explanation comes from how the tissues were isolated for scRNA-seq. When we performed scRNA-seq, we micro-dissected only the tumor tissue, in which there were very few microglia. However, the whole

brain with a tumor has much higher microglial populations (Please see the figure on the right). It has been shown that microglia are more prominent in the peritumoral area and sparsely distributed within the tumor bulk (PMID: 33766119). This is certainly a limitation of preclinical orthotopic tumor

implantation, as there is a substantial amount of microglia in human GBM (PMID: 32470396). Regardless of this limitation, following the reviewer's advice, we performed a microglia-centered GO analysis (Supplementary Fig. 7). The results indicate the microglia population was positively affected by our treatment to demonstrate a pro-inflammatory profile post-Combo therapy. Considering the efficiency of microglia to take up our nanoparticles (Fig. 3a) as the reviewer advised, these data may strongly suggest an important role of microglia in the antitumor effects of our therapy. Our subsequent studies need to be focused on the interplay between our nano-therapeutics and microglia in the context of glioma. We highly appreciate the reviewer's advice and have added these data, discussion, and limitations to the revised manuscript.

We would also like to thank the reviewer for the insights into the effect of empty B-LNP. Following the suggestion, we have evaluated the impact of diABZI-free B-LNP on the glioma immune microenvironment. Through flow cytometric analysis, we did not observe significant changes in TAMCs and lymphocytes in glioma-bearing mice treated with drug-free B-LNP (Supplementary Fig. 11). The drug-free B-LNP by itself only moderately improved the antitumor effect of radiotherapy in glioma-bearing animals (Supplementary Fig. 14). These results may highlight the importance of B-LNP-mediated coordinate modulation of multiple TAMC antitumor mechanisms, including promoting phagocytosis and activating STING pathways in TAMCs. These results have been included and briefly discussed in the revised manuscript.

Q5: Fig 4: the T cell analysis is convincing with the caveat that the DiABZI-free B-LNP condition is missing as well. In the caption of Fig 4a it should say 'B-LNP/DdiABZi programmed'...

A5: We would like to thank the reviewer for the expert insights. We have evaluated the effects of diABZI-free B-LNP on T cells by flow cytometry (Supplementary Fig. 11a). We did not observe significant changes in T cells in terms of abundance and activation by drug-free B-LNP. These results have been included and briefly discussed in the revised manuscript. We have also revised the caption of Fig 4a accordingly.

Q6: Fig 4i: a treatment scheme would help the reader (this was local application of antibody cocktails, I assume, should be highlighted in the figure).

A6: Following the reviewer's great suggestion, we have added the treatment scheme into **Fig. 4j** accordingly.

Q7: Fig 4j: it should say 'tumor-infiltrating'. I would update the scheme by adding the therapeutic modality applied.

A7: We would like to thank the reviewer for the suggestion and have revised the scheme accordingly.

Q8: Fig 5a: the condition B-LNP without diABZI is missing, it would be great to include if possible, as discussed above, or discuss as a limitation.

A8: We agree with the reviewer that it is a critical control to be involved. Following the advice, we have evaluated the therapeutic effect of diABZI-free B-LNP in glioma-bearing mice (**Supplementary Fig. 14**). Our data indicate a moderate improvement in the survival of CT-2A glioma-bearing mice receiving B-LNP treatments. These data highlight the essential role of STING activation and the consequent T cell antitumor activities in the anti-glioma efficacy of our therapy. These results are consistent with our observations in STING^{Gt} mice and Rag1^{-/-} mice in which the STING-related effects are absent (**Fig. 5b, c**).

Q9: Fig 6/systemic treatments: immunological effects of B-LNP/diABZI are clearly visible. Can the authors demonstrate homing of the nanoparticles to the tumor, by e.g. showing pRhodo Stained B-LNPs after iv injections in CT2A tumors in an additional figure? Any longitudinal/pharmacokinetic data would enhance the ms.

A9: We would like to thank the reviewer for the expert insights. Following the advice, we assessed the homing of nanoparticles to brain tumors over time by live animal imaging. Our results indicate that the intravenously injected Cy5.5-tagged nanoparticles were rapidly distributed to and accumulated in brain tumor sites (**Supplementary Fig. 16a, b**). A co-localization of nanoparticles and TAMCs further demonstrates the TAMC-targeting ability of nanoparticles in brain tumors by immunofluorescence staining (**Supplementary Fig. 16c**).

Q10: CAR T cell data and enhanced T-cell infiltration into the tumor after B-LNP/diABZI treatment: while the observed effect of enhanced T-cell homing is very interesting, the authors do not provide direct mechanistic clues on how this is working. Local proinflammatory cytokines certainly play a role, but the homing has to be induced in the periphery. Is there a chance to look at peripheral cytokines/chemokines that might explain the effect? Why was the CART cell experiment performed in RAG1^{-/-} mice and not in wildtype mice - please provide a short explanatory sentence.

A10: We would like to thank the reviewer for the expert insights. We have assessed the pro-inflammatory cytokine panel in serum and brain tumor samples using an MSD V-PLEX Cytokine Panel 1 Mouse Kit including CXCL10, CCL2, CCL3, CXCL2, IL-27p28/IL-30, IL33, IL15, IL9, and IL17A/F (**Fig. 4i, Supplementary Fig. 9**). Our results demonstrate that B-LNP/diABZI treatments increased a panel of pro-inflammatory cytokines in both serum (peripheral) and the brain tumor milieu (local), particularly T cell-recruiting/activating cytokines.

These data provide a mechanistic clue of the nanoparticle-stimulated brain tumor infiltration of T cells, including CAR T cells. For the proof of principle experiment, we tested our therapy in Rag^{-/-} mice to avoid immune-homeostatic competition (PMID: 35318469) and the potential lymphodepletion-induced cytokine production which might also affect T cell migration (PMID: 17255288).

Q11: Discussion: The term 'B-LNP/diABZI' should be used consistently, eg on line 332, 337 etc. Translational potential: I would not only focus on iv delivery (and, as mentioned above, tumor homing of B-LNP needs to be demonstrated). The effect seems higher when the drug is administered locally, which is also clinically feasible, and can be discussed as an adjuvant to radiation therapy or in case of recurrent GBM. Limitations as mentioned above need to be highlighted.

A11: We agree with the reviewer's thoughts and have revised the manuscript accordingly by checking the terms, discussing the potential use as an adjuvant to radiation therapy or in case of recurrent GBM, and highlighting the limitations.

Q12: Methods line 511: please provide the cell amount of CT2A and PF8 cells injected per mouse.

A12: We would like to thank the reviewer for the comment. We have added the cell amount into the methods accordingly.

We hope that all of the raised concerns have been addressed in a satisfactory manner and that this revised manuscript is now acceptable for publication in *Nature Communications*. Please let us know if we can provide more information.

Sincerely,

Maciej S. Lesniak MD, MHCM, FAANS

Michael J. Marchese Professor of Neurosurgery
Chairman, Department of Neurosurgery
Feinberg School of Medicine
Neurosurgeon-in-Chief, Northwestern Memorial
Hospital and Northwestern Medical Group
Program Leader, Neuro-Oncology
Lurie Cancer Center, Northwestern University

REVIEWERS' COMMENTS

Reviewer #1 (expert in nanoparticles in glioma, glioma therapy):

The authors have addressed all my comments satisfactorily. The data presented is novel and rigorous. The conclusions are in line with the results presented. This paper will be of high interest to the readership of Nature Communications.

Reviewer #2 (expert in CD47, macrophage signalling):

The authors have not addressed one prior comment from this reviewer. The prior comment, which should be addressed by the authors, is following.

“The CD47-SIRPα interaction is known to suppress the phagocytotic activity of phagocytes. Does B-LNP block the CD47-SIRPα interaction? The authors should show the significance of blockade of the CD47-SIRPα interaction by B-LNP in terms of its anti-tumorous effects on glioblastomas.”

Although B-LNP blocked the CD47 signals (Fig. 2e), the authors did not indicate that B-LNP blocked “the CD47-SIRPα interaction”. In addition, they did not evaluate “the significance of blockade of the CD47-SIRPα interaction by B-LNP” in terms of its anti-tumorous effects.

Reviewer #3 (expert in CD47, glioblastoma):

The authors thoroughly revised the manuscript and added sufficient evidence to support their conclusions. In my view, the manuscript is suitable for publication in Nature Communications, and showcases an important novel therapeutic modality against glioblastoma.